# A Histidine pH sensor regulates activation of the Ras-specific guanine nucleotide exchange factor RasGRP1

Yvonne Vercoulen[1,2†], Yasushi Kondo[3,4†], Jeffrey S Iwig[3,4†‡], Axel B Janssen[1], Katharine A White[5], Mojtaba Amini[2], Diane L Barber[5], John Kuriyan[3,4,6,7,8]\*, Jeroen P Roose[1]\*

[1]Department of Anatomy, University of California, San Francisco, San Francisco, United States; [2]Molecular Cancer Research, Center for Molecular Medicine, UMC Utrecht, Utrecht University, Utrecht, Netherlands; [3]Department of Molecular and Cell Biology and Chemistry, University of California, Berkeley, United States; [4]California Institute for Quantitative Biosciences, University of California, Berkeley, United States; [5]Department of Cell and Tissue Biology, University of California, San Francisco, San Francisco, United States; [6]Howard Hughes Medical Institute, University of California, Berkeley, United States; [7]Department of Chemistry, University of California, Berkeley, United States; [8]Divisions of Molecular Biophysics and Integrated Bioimaging, Lawrence Berkeley National Laboratory, Berkeley, United States

\*For correspondence:
kuriyan@berkeley.edu (JK);
jeroen.roose@ucsf.edu (JPR)

[†]These authors contributed equally to this work

Present address: [‡]Carmot Therapeutics, San Francisco, United States

**Abstract** RasGRPs are guanine nucleotide exchange factors that are specific for Ras or Rap, and are important regulators of cellular signaling. Aberrant expression or mutation of RasGRPs results in disease. An analysis of RasGRP1 SNP variants led to the conclusion that the charge of His 212 in RasGRP1 alters signaling activity and plasma membrane recruitment, indicating that His 212 is a pH sensor that alters the balance between the inactive and active forms of RasGRP1. To understand the structural basis for this effect we compared the structure of autoinhibited RasGRP1, determined previously, to those of active RasGRP4:H-Ras and RasGRP2:Rap1b complexes. The transition from the autoinhibited to the active form of RasGRP1 involves the rearrangement of an inter-domain linker that displaces inhibitory inter-domain interactions. His 212 is located at the fulcrum of these conformational changes, and structural features in its vicinity are consistent with its function as a pH-dependent switch.

DOI: https://doi.org/10.7554/eLife.29002.001

## Introduction

The Ras family of small G-proteins, including Ras and Rap, are molecular switches that transmit signals when bound to GTP (*Rojas et al., 2012*). Ras-family members are activated by specific guanine-nucleotide exchange factors (that we term GEFs for simplicity here), such as Son of Sevenless, that activates Ras (*Boriack-Sjodin et al., 1998*), and Epac1 and Epac2, that activate Rap (*Rehmann et al., 2006*). GEFs activate Ras-family members by triggering the release of GDP and its replacement by GTP (*Wittinghofer and Vetter, 2011*). These GEFs, of which there are several distinct families in humans, activate Ras-family members in response to a variety of signals, and thereby initiate downstream kinase signaling cascades (*Bos et al., 2007*; *Cherfils and Zeghouf, 2013*). One family of these GEFs is comprised of the Ras guanine-nucleotide releasing proteins (RasGRPs) that can activate Ras or Rap, of which there are four in humans. RasGRP1 is most extensively studied in T

**eLife digest** Complex chain reactions between many kinds of molecules regulate every process in the body. For example, the signaling molecule Ras helps the cell to grow and divide. However, abnormally high levels of Ras signals can cause cancer.

Ras is activated by proteins called exchange factors. One of the families of Ras exchange factors – RasGRP – plays important roles in immune and bleeding disorders, and certain cancers. In 2013, researchers studied the structure of one of these exchange factors, called RasGRP1, while it was inactivated. Inactive RasGRP proteins have a 'closed' structure, which must 'open up' when they are activated.

Vercoulen, Kondo, Iwig et al. – who include several of the researchers involved in the 2013 study – have now investigated the regulation of various RasGRP family members. The protein structures of the active forms of two family members were determined and compared with the structure of inactive RasGRP1. In parallel, Vercoulen et al. analyzed how genetic mutations that alter some of the amino acids that make up RasGRP1 affect Ras signaling in cells. This revealed that a particular amino acid, histidine 212, plays a key role in activating RasGRP1.

Histidines can be in a positively charged or neutral form depending on other surrounding amino acids and on the acidity of the cell's interior. The interiors of cells that receive an external signal often decrease in acidity, and cancer cells tend to have less acidic interiors than normal cells. Vercoulen et al. reveal that a change in the charge on histidine 212 from positive to neutral opens up the RasGRP1 protein. Histidine 212 therefore acts as an acidity sensor that activates RasGRP1 when the inside of the cell becomes less acidic as external signals are received.

Since RasGRP proteins play important roles in many diseases, understanding how cell acidity regulates RasGRPs has wide medical relevance. In the future, the protein structures of the RasGRP family members and the method developed in this study could be used to explore how they contribute to disease.

DOI: https://doi.org/10.7554/eLife.29002.002

lymphocytes (*Dower et al., 2000*; *Priatel et al., 2002*; *Roose et al., 2005*; *Daley et al., 2013*), RasGRP2 in neutrophils and platelets (*Lozano et al., 2016*; *Stone, 2011*), RasGRP3 in B lymphocytes (*Teixeira et al., 2003*), and RasGRP4 in mast cells (*Yang et al., 2003*). However, RasGRPs can be found in other cell types as well, including epithelial lineages (*Depeille et al., 2015*). T cell receptor (TCR) or B cell receptor (BCR) stimulation leads to an increase of diacylglycerol at the membrane, protein kinase C (PKC) activation, and increases in intracellular calcium levels (*Myers and Roose, 2016*). RasGRP1 is recruited to the plasma membrane, where it binds diacylglycerol to enable inter-action with Ras. Upon receptor stimulation, RasGRP1 is phosphorylated by PKC (*Zheng et al., 2005*; *Roose et al., 2007*), which enhances RasGRP1's GEF activity through unknown mechanisms. RasGRP1 contains a calcium-binding EF hand domain and calcium binding to RasGRP1 induces allo-steric changes that release autoinhibition (*Iwig et al., 2013*).

Dysregulated RasGRPs can cause aberrant signaling and result in disease. Alterations in RasGRP1 expression contribute to diseases such as skin carcinoma (*Diez et al., 2009*; *Luke et al., 2007*), colo-rectal cancer (*Depeille et al., 2015*), and leukemia (*Ksionda et al., 2016*; *Hartzell et al., 2013*). Ele-vated RasGRP3 expression has been reported in breast cancer (*Nagy et al., 2014*), prostate cancer (*Yang et al., 2010*), lymphoma (*Teixeira et al., 2003*), cutaneous melanoma (*Yang et al., 2011*), and uveal melanoma (*Chen et al., 2017*). A polymorphism (Arg 519 Gly) in *Rasgrp1* in mice results in T cell abnormalities and autoimmunity (*Daley et al., 2013*). Furthermore, several genetics studies have linked single nucleotide polymorphisms (SNPs) in *RasGRP1* to human autoimmune disease (*Plagnol et al., 2011*; *Qu et al., 2009*), and low RasGRP1 levels have been detected in T lympho-cytes from patients with Systemic Lupus Erythematosus (SLE) (*Yasuda et al., 2007*), and rheumatoid arthritis (RA) (*Golinski et al., 2015*). Complete RasGRP1 deficiency in a patient leads to a novel pri-mary immunodeficiency, with impaired activation and proliferation of the patient's T- and B- cells and defective killing by cytotoxic T cells and NK cells (*Roose, 2016*; *Salzer et al., 2016*). *Rasgrp2* deficiency in mice results in excessive bleeding, caused by defective platelets aggregation and degranulation (*Crittenden et al., 2004*). Moreover, polymorphisms in *RasGRP2*, either converting

Arg 113 into a stop codon, or missense mutations (Gly 248 Trp or Ser 381 Phe), cause a platelet disorder in patients (*Lozano et al., 2016*;*Canault et al., 2014*).

The N-terminal portion of the RasGRPs contains the catalytic module that is common to other GEFs that operate on Ras-family members (*Figure 1A*). This module consists of a Ras-exchanger motif (REM) domain followed by a Cdc25 domain (*Boguski and McCormick, 1993*;*Fam et al., 1997*). As first revealed by structural analysis of the Ras-specific GEF Son-of-Sevenless (SOS), the Cdc25 domain interacts with Ras and is responsible for nucleotide release, and the REM domain provides structural support for the Cdc25 domain (*Boriack-Sjodin et al., 1998*). The remaining portion of the RasGRP proteins consists of a $Ca^{2+}$-binding EF domain and a C1 domain that binds diacylglycerol for membrane localization (*Beaulieu et al., 2007*; *Ebinu et al., 1998*; *Zahedi et al., 2011*). RasGRP1 has a C-terminal coiled-coil domain that is missing in the other family members. The common set of domains in RasGRP-1, -2, -3, and -4 show a high degree of sequence conservation (*Ksionda et al., 2013*).

A nearly complete understanding of the regulation of SOS has been provided through structural and functional studies (reviewed in *Jun et al., 2013*); The exchange-factor activity of the catalytic module of SOS is inhibited by the action of the N- and C-terminal segments (*Sondermann et al., 2004*), and the activation of SOS requires allosteric feedback from Ras•GTP binding to a site that is distal to the catalytic site where nucleotide is exchanged (*Roose et al., 2007*; *Margarit et al., 2003*; *Boykevisch et al., 2006*). Once activated, SOS requires multiple plasma membrane-anchoring mechanisms to signal efficiently to Ras (*Findlay et al., 2013*; *Christensen et al., 2016*), and SOS signaling is terminated in part via endocytosis (*Christensen et al., 2016*). The regulation of the Rap-specific GEF Epac2 is also well understood (*Rehmann et al., 2006*; *Rehmann et al., 2008*).

Much less is known about the regulation and activation of any of the four RasGRP proteins. We had previously analyzed the structural basis for the autoinhibition of RasGRP1, which is very different from that of SOS (*Figure 1B*). In SOS, the active site is open in the autoinhibited form, and inactivation appears principally to be the result of the blockage of allosteric Ras binding. Allosteric Ras binding causes a conformational change at the active site in SOS. In contrast, in RasGRP1, the active site is blocked by the linker connecting the Cdc25 domain to the EF domain (the Cdc25-EF linker) (*Iwig et al., 2013*). In addition, we proposed that dimerization of RasGRP1 through the C-terminal coiled-coil domain results in blockage of the membrane-interacting surface of the C1 domain. In the structure of autoinhibited RasGRP1, the EF and C1 domains are docked on the base of the Cdc25 domain, and this interaction steers the Cdc25-EF linker through the Ras-binding site of the Cdc25 domain (*Iwig et al., 2013*). Calcium binding to the EF-hand of RasGRP1 promotes a conformational change that is likely to result in displacement of the Cdc25-EF linker from the catalytic site (*Iwig et al., 2013*). Thus, the EF domain plays both autoinhibitory and activating roles in regulating RasGRP1, and a point mutation in the EF domain leads to an autoimmune phenotype in mice (*Daley et al., 2013*). Sequence comparisons suggest that this autoinhibitory mechanism is conserved in the three other RasGRP proteins.

By scanning and testing human single nucleotide variants (SNVs) from genome databases, we uncovered that a conserved histidine residue in the Cdc25 domain of RasGRP1 (His 212) is critical for autoinhibition. We show that the activity of RasGRP1 is sensitive to cellular pH, and that His 212 is critical for this pH sensitivity. Stimulation of lymphocytes results in an increase in intracellular pH (pHi) (*Cheung et al., 1988*; *Fischer et al., 1988*;*Mills et al., 1985*), which can lead to deprotonation and conversion of histidine residues from positively charged to neutral (*Schönichen et al., 2013*). We find that increasing pHi synergizes with receptor stimulation to activate the cellular RasGRP1-Ras-ERK pathway. Conversely, replacing His 212 by a positively charged lysine residue (H212K) prevents the recruitment of RasGRP1 to the membrane, presumably by stabilizing the autoinhibited form.

In order to understand the role of His 212 in regulation of RasGRP1 we sought to compare the structure of autoinhibited RasGRP1, determined previously (*Iwig et al., 2013*), to that of the active form in complex with Ras. We have been unable to crystallize a RasGRP1:Ras complex. Instead, we determined the structures of the catalytic modules of two other members of the family, RasGRP4 and RasGRP2, bound to nucleotide-free HRas and Rap1B, respectively. These structures reveal a key role for the REM-Cdc25 linker in determining the switch from the autoinhibited to the active states. The structure of this linker is fully resolved in the RasGRP2:Rap1B complex, and the linker conformation is incompatible with the inhibitory docking of the EF domain on the Cdc25 domain. The

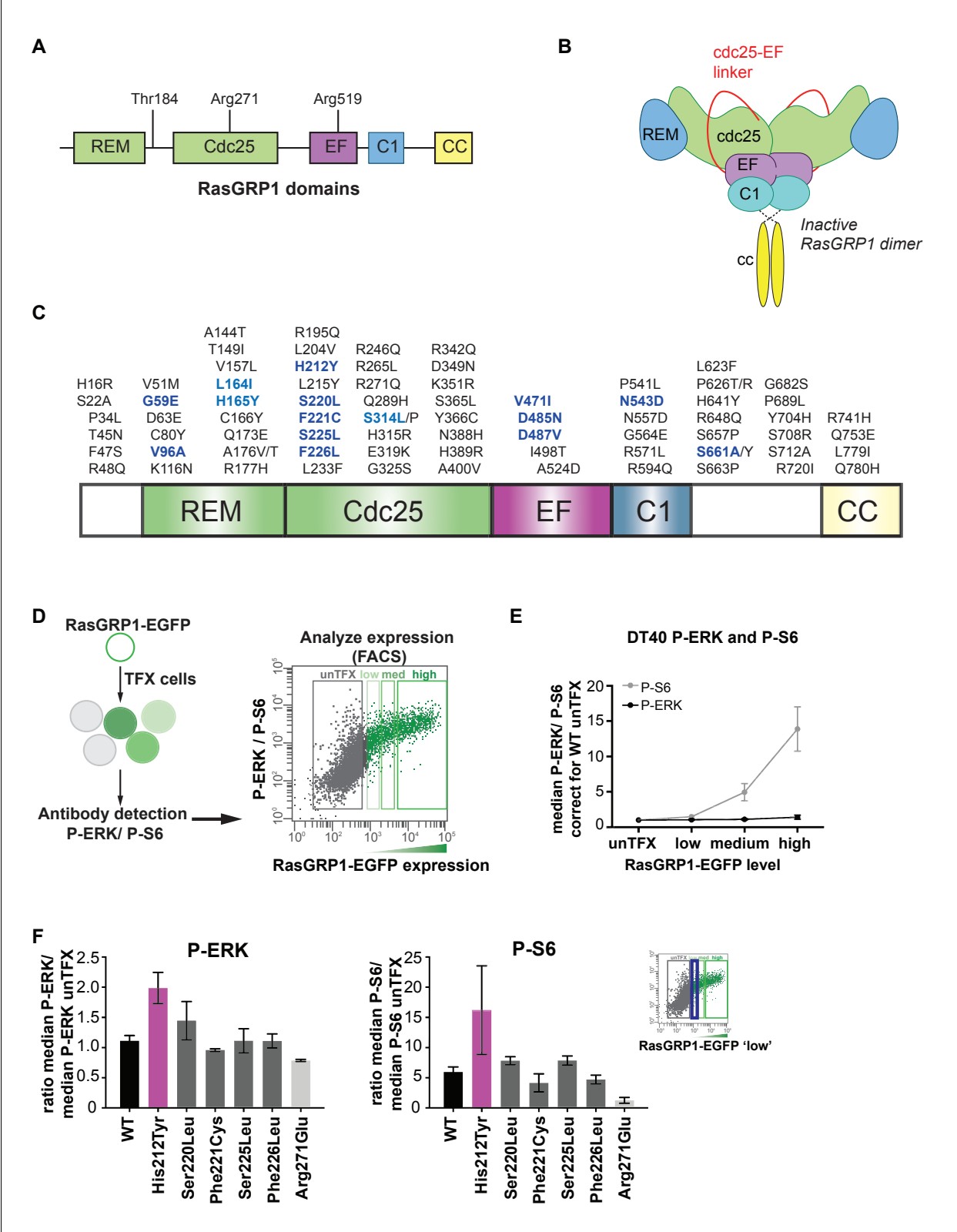

**Figure 1.** Altered basal RasGRP1 signaling by a human His 212 Tyr SNV. (**A**) The REM, Cdc25, EF hands, C1 and coiled-coil are known protein domains of RasGRP1. Indicated are residues in different domains that have been shown to impact RasGRP1 activity (Thr 184, Arg 271, Arg 519). (**B**) Schematic of an autoinhibited RasGRP1 homodimer. (**C**) Alignment of SNVs (missense variants) are in the coding regions of RasGRP1 collected from online repositories dbSNP, COSMIC, NCBI in 2012 and 2014. (**D**) Assay to determine RasGRP1 activity in cells. RasGRP1⁻/⁻RasGRP3⁻/⁻ DT40 cells are

*Figure 1 continued on next page*

Figure 1 continued

transfected with EGFP-RasGRP1 wildtype or mutant plasmid. Intracellular phosphorylated-ERK (P-ERK) and phosphorylated ribosomal protein S6 (P–S6) are detected by flow cytometry that allows for analysis of cells expressing different levels of RasGRP1-EGFP. (E) Analysis of basal S6- and ERK-phosphorylation as a function of RasGRP1 expression levels (N = 7, unTFX = untransfected). (F) Effect of SNVs on basal S6- and ERK- phosphorylation. Graphs display RasGRP1-EGFP low expressing cells (N = 2). (E,F) Y-axes show the average ratios of median fluorescent phospho-protein signals/median fluorescent phospho-protein signals from untransfected cells (baseline). Shown are means ±SD.

DOI: https://doi.org/10.7554/eLife.29002.003

sequence of the REM-Cdc25 linker is conserved between RasGRP1 and RasGRP2, and we infer that the activation of the RasGRP1 involves correlated movements in the REM-Cdc25 linker and the Cdc25-EF linker. The structures show that the location of His 212 is such that the charge on this residue could alter the balance between the active and inactive states of RasGRP1.

## Results and discussion

### Human missense variants of RasGRP1

Single nucleotide variants that cause amino acid substitutions (missense variants; SNVs) are frequent and generate human genetic variation: most people inherit ~12,000 missense gene variants (*Abecasis et al., 2010*). We identified all SNVs reported for RasGRP1 in publicly available databases, and took a shotgun approach to test a panel of these SNVs for their ability to alter RasGRP1 regulation (*Figure 1C*).

We have not been able to express full length RasGRP1 and study the autoinhibitory mechanisms in cell-free assays (*Iwig et al., 2013*). Therefore, to assess the potential signaling impact of these SNVs, we used RasGRP1$^{-/-}$RasGRP3$^{-/-}$ DT40 cells (a chicken B cell line) genetically deleted for *RasGRP1* and *RasGRP3* and reconstituted these cells via transfection with wildtype EGFP-RasGRP1 (WT) or a catalytically inactive RasGRP1 (Arg271Glu) as before (*Iwig et al., 2013*), or with a panel of RasGRP1 SNVs (indicated in bold and blue in *Figure 1C*). This assay allows for activity assessment of RasGRP1-ERK signaling (*Iwig et al., 2013*), but also of RasGRP1-mTORC1-p70S6 kinase signaling resulting in phosphorylation of ribosomal protein S6 (P-S6). Precisely how RasGRP1 signals to the S6 pathway is still unresolved and is not the focus of this study here, but the Arg 519 Gly mutation in *Rasgrp1*$^{Anaef}$ mice results in higher basal S6 signaling, T cell abnormalities, and autoimmunity (*Daley et al., 2013*). To assess the basal activity of RasGRP1 and its SNVs, we used quantitative flow cytometric analysis of phosphorylated ERK (P-ERK) and phosphorylated ribosomal protein S6 (P-S6) levels as a function of the expression level of RasGRP1-EGFP (*Figure 1D*). Our quantitative flow cytometric analyses revealed that RasGRP1 signals strongly to P-S6 in the basal state; basal signals from RasGRP1 to ERK do occur, but are more modest (*Figure 1E*).

Most SNVs were neutral, with signaling features either similar to WT RasGRP1 or with lower activity. There are numerous possible reasons for SNVs signaling at lower strength, including reduction in protein stability (data not shown). However, the His 212 Tyr SNV signaled more strongly to ERK than WT, indicating altered regulation of RasGRP1. More detailed analysis of the cellular biochemical traits of the SNVs in the His 212 region demonstrated that His 212 Tyr, but not Ser 220 Leu, Phe 221 Cys, and Phe 226 Leu, resulted in increased basal signals to P-ERK and P-S6 in unstimulated cells as compared to WT RasGRP1 (*Figure 1F*).

### His 212 controls basal RasGRP1 signals

His 212 in RasGRP1 is conserved among all vertebrate RasGRPs, and is present in most RasGRP proteins from lower organisms (*Figure 2A*). This residue is located in the first helix of the Cdc25 domain, and is far from the Ras-binding site. We assessed the activity of RasGRP1 bearing mutations at position 212 in transfected cells. Analysis of the human SNV variant His 212 Tyr and His 212 Ala, both alterations to neutral residues, showed increased basal signals to P-ERK in RasGRP1$^{-/-}$RasGRP3$^{-/-}$ DT40 cells (*Figure 2B*, *Figure 2—figure supplement 1A*) as well as in JPRM441 (*Figure 2C*, *Figure 2—figure supplement 1B*), a RasGRP1-deficient Jurkat T cell leukemia line that we previously exploited to asses RasGRP1 function (*Roose et al., 2005*; *Iwig et al., 2013*). Similarly, the His 212 Tyr and His 212 Ala variants of RasGRP1 signaled stronger to P-S6 in the DT40 cell system

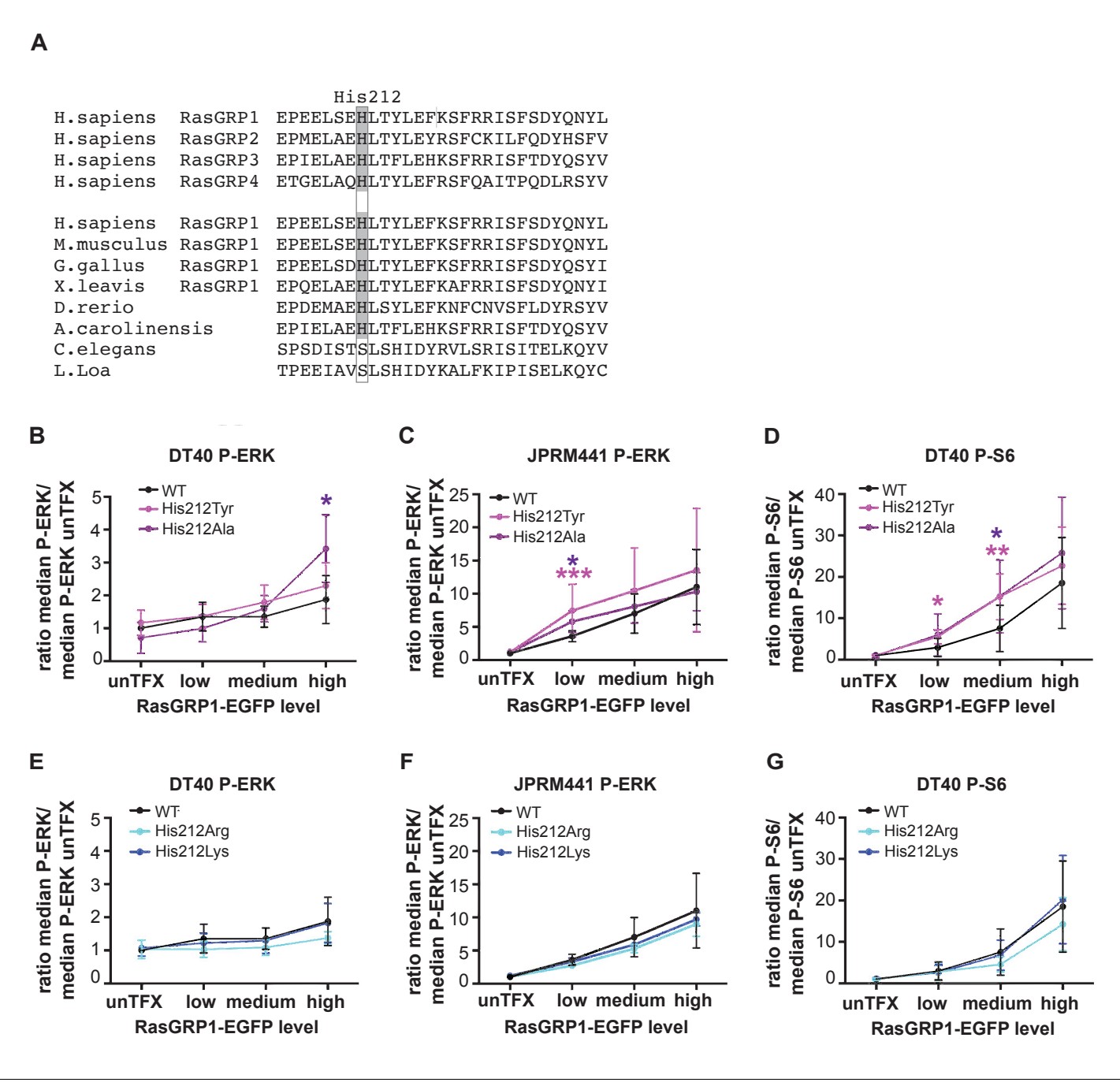

**Figure 2.** A neutral charge on Histidine 212 in RasGRP1 enhanced basal activity. (**A**) Alignment of RasGRP isoforms and RasGRP1 sequences in different species to determine conserved residues. His 212 conservation is indicated by grey color. (**B–G**) Low RasGRP1-expressing Jurkat cells (JPRM441), or RasGRP1$^{-/-}$RasGRP3$^{-/-}$ DT40 cells were transfected with wildtype- or variants of EGFP-RasGRP1. (**B–D**) His 212 was mutated into neutrally charged residues (His 212 Ala, His 212 Tyr), or (**E–G**) into positively charged residues (His 212 Arg, His 212 Lys). RasGRP1 signaling was determined by levels of phosphorylated ERK (P-ERK) and S6 (P–S6). Shown are average median levels of P-ERK and P-S6, corrected for untransfected cells for each experiment, to prevent experimental fluctuation in fluorescent signal. (**Figure 2B and E**: WT N = 8, His 212 Tyr N = 8, His 212 Ala N = 6, His 212 Arg N = 3, His 212 Lys N = 3, **Figure 2C and F**: WT N = 10, His 212 Tyr N = 7, His 212 Ala N = 7, **Figure 2D and G**: WT, His 212 Arg, His 212 Lys, all N = 6). Kruskal-wallis test was used, with post-test Dunn's multiple comparisons, comparing each mutant to WT (*p<0.05, **p<0.01, ***p<0.001).

DOI: https://doi.org/10.7554/eLife.29002.004

The following figure supplement is available for figure 2:

*Figure 2 continued on next page*

*Figure 2 continued*

**Figure supplement 1.** Analyses of RasGRP1 level-dependent effects on ERK- and S6-kinase pathway activation.

DOI: https://doi.org/10.7554/eLife.29002.005

(*Figure 2D*, *Figure 2—figure supplement 1C*); PTEN- and SHIP1-deficiency in Jurkat results in hyperactive PI3kinase signals (*Abraham and Weiss, 2004*), making analysis of PI3K-dependent signals, such as those to S6, difficult in Jurkat. Thus, the His 212 Tyr and His 212 Ala variants are less autoinhibited. Mutation of His 212 to positively charged residues, i.e. His 212 Arg and His 212 Lys, maintained autoinhibition of RasGRP1 activity in unstimulated cells with a similar efficiency as WT RasGRP1 (*Figures 2E,F and G*, *Figure 2—figure supplement 1D,E and F*). Note that our analysis of the effects of these, and other, mutations in RasGRP1 is necessarily restricted to cell-based assays. As shown previously, the RasGRP1 construct used here is released from autoinhibition when studied in vitro, in solution (*Iwig et al., 2013*). The inability of in vitro measurements to capture details of the regulatory mechanism may reflect the role of dimerization in maintaining autoinhibition – the construct used in these studies lacks the C-terminal dimerization domain, and we have not yet succeeded in expressing and purifying full-length RasGRP1.

## The charge of His 212 impacts activation of RasGRP1

The same transfection phospho-flow strategy also allows for evaluation of RasGRP1 function following B cell receptor (BCR) stimulation, by analyzing kinase pathway signals in cells that express low levels of RasGRP1-EGFP, which we gate on. Upon BCR stimulation with M4 antibody, we typically observe induced levels of P-ERK and P-S6 that are 7–9 fold and 2–3 fold over baseline, respectively (*Figure 3A*). Note that induction of ERK phosphorylation is more robust than S6 phosphorylation upon BCR stimulation. Thus, S6 phosphorylation appears to be a more robust event in the basal state (*Figure 1*) and ERK phosphorylation is a more robustly triggered kinase pathway in stimulated cells. To understand the role of His 212 in the activation of RasGRP1, we used our transfection phospho-flow platform and BCR stimulation.

Changing His 212 to neutral residues, as in the human SNV His 212 Tyr or the designed His 212 Ala variant, resulted in increased levels of BCR-induced phosphorylation of ERK and S6 (*Figure 3B and C*). Conversely, altering His 212 to positively charged residues, either His 212 Arg or His 212 Lys, impaired the BCR-induced ERK and S6 responses (*Figure 3D and E*). The transfection platform with EGFP-tagged RasGRP1 also allowed us to investigate the recruitment of RasGRP1 to the plasma membrane (*Bivona et al., 2003*; *Daley et al., 2013*). Wildtype and His 212 Tyr versions of RasGRP1-EGFP revealed very similar efficiency of plasma membrane recruitment upon BCR-stimulation (*Figure 3F and G*, and *Figure 3—figure supplement 1*). Most striking was the complete lack of RasGRP1 plasma membrane recruitment for the positively charged mutant His 212 Lys (*Figure 3G* and *Figure 3—figure supplement 1*). This is remarkable because BCR-stimulation generates a robust intracellular calcium flux and increased levels of DAG (*Myers and Roose, 2016*), which recruits RasGRP1 to the membrane and activates it. Thus, the His 212 Lys mutation blocks the responsiveness of RasGRP1 to these calcium and DAG cues. Together, these data suggest that the positive charge of His 212 contributes to RasGRP1 autoinhibition in the basal state (*Figure 2*), and that a neutral residue at position 212 is essential to allow for RasGRP1 plasma membrane recruitment and signal output to the ERK and S6 pathways (*Figure 3*).

## His 212 functions as a pH sensor that modulates RasGRP1 activity

Given that increases in intracellular pH can result in deprotonation of histidines (*Schönichen et al., 2013*) (*Figure 4A*), we next explored whether RasGRP1 is regulated by intracellular pH through His 212. Cells in an activated state often display increases in intracellular pH (pHi); For example, the pHi is elevated in tumor cells, which promotes tumor survival and progression (*Webb et al., 2011*; *Grillo-Hill et al., 2015*). Stimulation of DT40 cells by PMA (a diacylglycerol analogue) and ionomycin (calcium ionophore), mimicking lymphocyte receptor signaling events that connect to RasGRP1, resulted in a consistent increase of the intracellular pH (*Figure 4B*).

Intracellular pH can be experimentally changed by regulating the activity of plasma membrane $H^+$ and $HCO_3^-$ ion transporters. Additionally, treating cells with low concentrations (10–20 mM) of

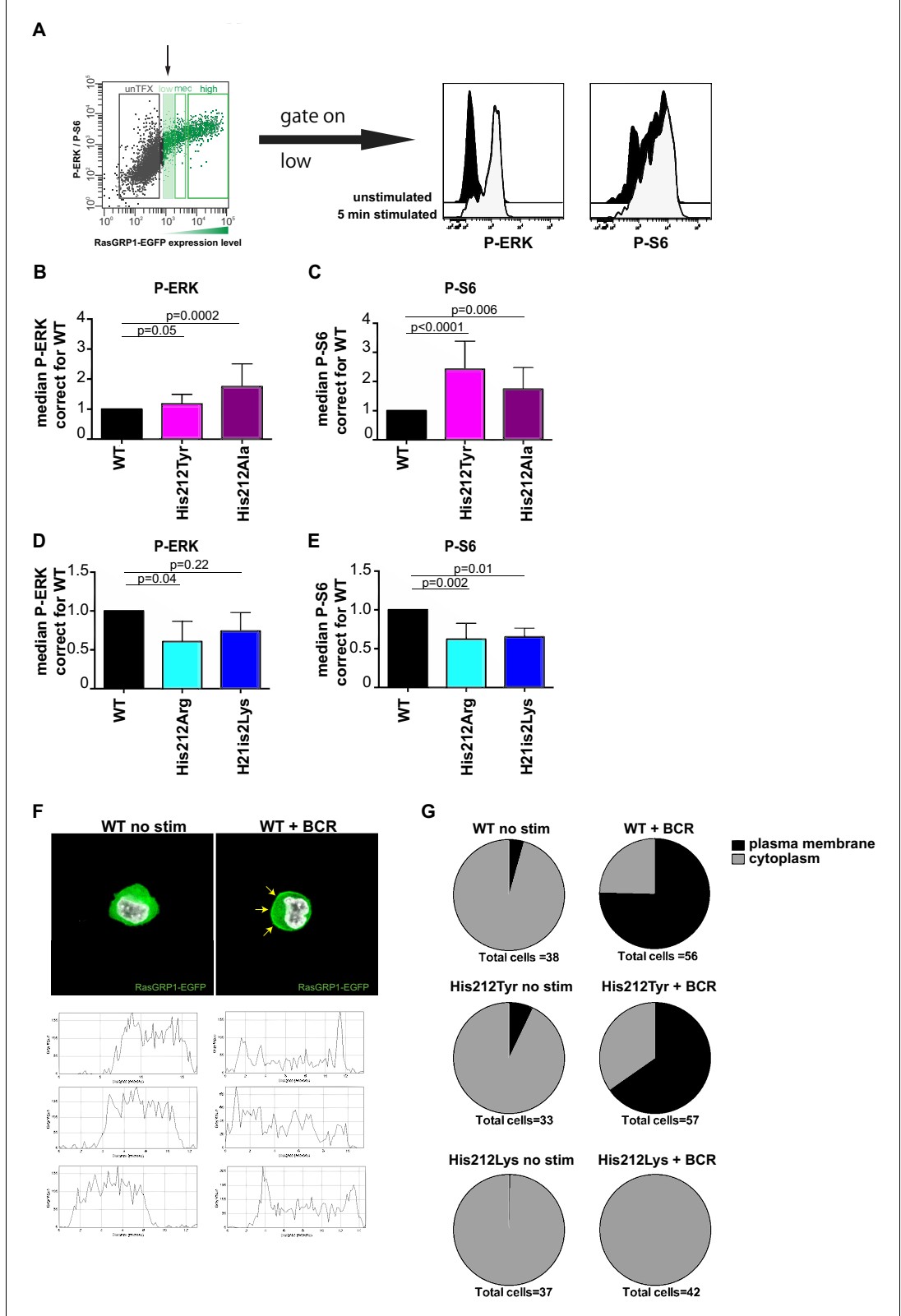

**Figure 3.** A constitutive positive charge on His 212 in RasGRP1 reduces kinase signaling and prevents RasGRP1 plasma membrane recruitment. (**A**) Assay to determine RasGRP1 activity in receptor stimulated cells using RasGRP1-/-RasGRP3-/- DT40 cells, BCR stimulation, and gating on low levels of transfected RasGRP1-EGFP. (**B–E**) Effects of H212 mutations on RasGRP1 activity were tested upon B cell receptor (BCR) stimulation. Shown are average median levels of P-ERK and P-S6 in EGFP-low expressing cells, corrected for wildtype stimulated cells. WT was set on 1.0. Kruskal-wallis test (*Figure 3C*

*Figure 3 continued on next page*

*Figure 3 continued*

*and D*; WT N = 10, His 212 Tyr/Ala N = 7, *Figure 3D and E*; all N = 6) was used, with post-test Dunn's multiple comparisons, comparing each mutant to WT. P-values are shown. All graphs show mean ±SD. (F) Localization of full length expressed RasGRP1-EGFP in DT40 RasGRP1$^{-/-}$ RasGRP3$^{-/-}$ cells, showing basal state (no stim, cytoplasmic), and upon B cell receptor stimulation (+BCR, arrows indicate plasma membrane localization). Below, representative profiles of expression are shown, peaks on the left and/or right side indicate membrane localization. (G) Membrane localization of WT, and mutants His 212 Tyr, His 212 Lys RasGRP1 was determined by microscopy analysis of unstimulated and BCR stimulated cells. Pie charts depict average individual cell counts analyzed in 3 separate experiments. Total number of cells analyzed under each condition are listed below charts.
DOI: https://doi.org/10.7554/eLife.29002.006

The following figure supplement is available for figure 3:

**Figure supplement 1.** Analyses of cellular RasGRP1 localization as a function of His 212 variation.
DOI: https://doi.org/10.7554/eLife.29002.007

ammonium chloride (NH$_4$Cl) increases pHi, and pulsing with NH$_4$Cl results in a substantially lower pH with NH$_4$Cl removal. Decreased pHi after pulsing is retained in the presence of EIPA (a specific inhibitor of the Na$^+$/H$^+$ exchanger NHE1) by blocking H$^+$ efflux (*Figure 4C*) (*Nakanishi et al., 1992*). In our transfected RasGRP1$^{-/-}$RasGRP3$^{-/-}$ DT40 cells, treating with 15 mM NH$_4$Cl increased pHi, whereas a NH$_4$Cl pulse followed by EIPA treatment resulted in a trend towards decreased pHi (*Figure 4D*).

We chose NH$_4$Cl and EIPA to modulate the pHi so that we could subsequently assess how this may impact RasGRP1 activity following BCR stimulation of cells. As such, we treated cells with NH$_4$Cl or NH$_4$Cl + EIPA prior to BCR receptor stimulation. NH$_4$Cl pretreatment was compared to vehicle pretreatment, and resulted in augmentation of the BCR-induced ERK phosphorylation in RasGRP1$^{-/-}$RasGRP3$^{-/-}$ DT40 cells that were transfected with WT RasGRP1 (*Figure 4E*). In contrast, NH$_4$Cl + EIPA treatment did not increase BCR-induced ERK signaling, but resulted in a trend towards decreased ERK signals (*Figure 4F*). It should be noted that lowering of pHi has been reported to activate cell death pathways (*Lagadic-Gossmann et al., 2004*). However, in our experiments the value of pHi never reached levels low enough to induce cell death effects that are observed under some conditions when pHi <7.0 (*Matsuyama et al., 2000*).

With these approaches, we investigated if the augmentation of BCR-induced ERK signals at increased pHi was the result of deprotonation of His 212 in RasGRP1 expressed in these cells. We used untransfected cells or RasGRP1$^{-/-}$RasGRP3$^{-/-}$ DT40 cells transfected with WT RasGRP1, His 212 Tyr, or His 212 Lys. First, we established that all four cell populations demonstrated very similar pHi increases or decreases upon NH$_4$Cl incubation or a NH$_4$Cl pulse with EIPA, respectively (*Figure 4G*). We found that increased pHi (*Figure 4H*) or decreased pHi (*Figure 4I*) only impacted the level of ERK signaling when His 212 in RasGRP1 was intact. His 212 Tyr and His 212 Lys lost the pHi-dependent regulation. These data indicate that His 212 in RasGRP1 is a pH sensor, and that increases in intracellular pH and His 212 deprotonation promote RasGRP1 activation.

## Structures of active complexes of RasGRP2 and RasGRP4

In order to understand the role of His 212 in controlling the activation state of the protein, we wished to determine the structure of RasGRP1 bound to Ras, i.e., in an active form, but we failed to crystallize this complex. In order to obtain structures of active complexes we shifted our focus to other RasGRP family members, since they are closely related in sequence (*Figure 5A* and *Figure 5— figure supplement 1*). The different RasGRPs have been reported to prefer different members of the Ras subfamily, based on gene knockout experiments (see review [*Ksionda et al., 2013*]). We first determined the specificity of all four RasGRP family members for Ras and Rap, using an in vitro GEF assay and only the catalytic domains of RasGRPs. We found that RasGRP1 and RasGRP4 are specific for Ras, whereas RasGRP2 has a clear preference for Rap1B. RasGRP3 works equally efficiently as a GEF for Ras, Rap1B and Rap2A (*Figure 5B*). These findings are in agreement with published work that compared a large panel of GEFs and their preference for specific small GTPases as substrates (*Popovic et al., 2013*).

We were successful in crystallizing the catalytic modules of RasGRP4 bound to HRas and RasGRP2 bound to Rap1B (see Materials and methods for details). Comparison of the structures of the RasGRP4:HRas and RasGRP2:Rap1B complexes show that nucleotide-free HRas and Rap1B are bound in a similar fashion to the two RasGRP proteins (*Figure 5C and D*), and that the binding

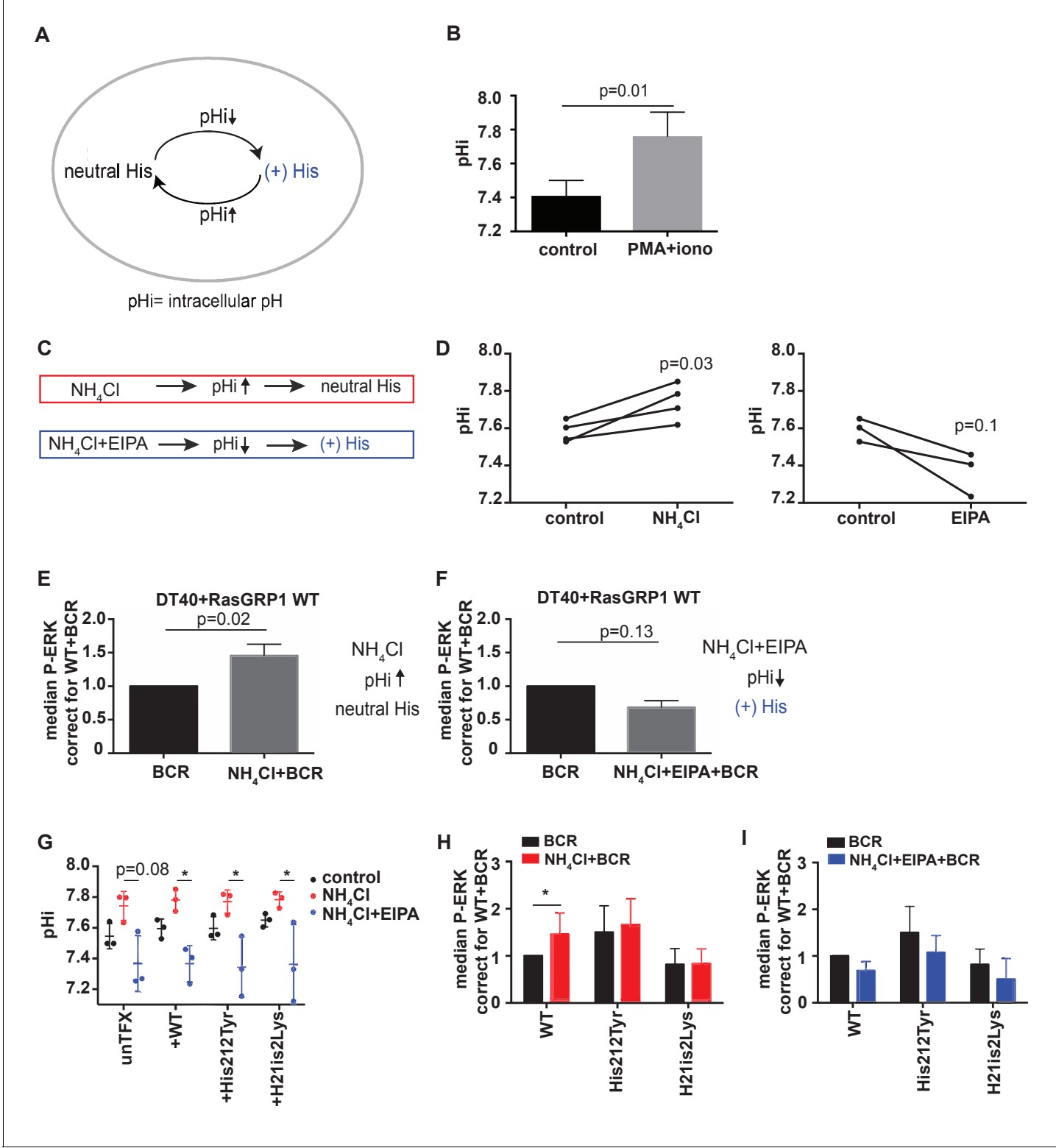

**Figure 4.** His 212 in RasGRP1 is a pH sensor.  (**A**) Intracellular pH (pHi) influences the charge of histidines (His). Low pHi shifts equilibrium towards protonated histidine (His+), while in high pHi conditions equilibrium shifts towards deprotonated histidine (His neutral). (**B**) Intracellular pH (pHi) was measured upon PMA +ionomycin stimulation in DT40 cells. Shown are means ±SD (N = 3). Paired T-test was performed, p-value is shown. (**C**) NH4Cl (NH$_4$Cl) increases pHi, resulting in a neutral Histidine, while NH4Cl loading and washing, followed by EIPA (NH$_4$Cl + EIPA) decreases pHi leading to a protonated Histidine. H212Y mimics neutrally charged histidine, and H212K mimics protonated histidine. (**D**) DT40 RasGRP1$^{-/-}$RasGRP3$^{-/-}$ cells

*Figure 4 continued on next page*

*Figure 4 continued*

transfected with wildtype (WT) RasGRP1-EGFP were treated with NH$_4$Cl (N = 4) or with NH$_4$Cl + EIPA (N = 3). pHi was determined in triplicates of controls and treated cells. Individual experiments are shown. Paired T-test was performed. (E, F) NH$_4$Cl, NH$_4$Cl + EIPA, and control treatment was followed by B cell receptor (BCR) stimulation. Shown are average median levels ± SD of P-ERK in EGFP-low expressing cells, corrected for wildtype BCR stimulated cells. WT was set on 1.0. (BCR, NH$_4$Cl: N = 7, NH$_4$ClL + EIPA: N = 4). Wilcoxon test was performed. (G) pHi was determined in untransfected and cells transfected with WT or mutant RasGRP1-EGFP. Shown are means ±SD (N = 3). Kruskal-Wallis tests were performed with post-test Dunn's multiple comparisons between control, NH$_4$Cl and NH$_4$Cl + EIPA for each unTFX, or transfected RasGRP1 WT or mutant. P-values are depicted for p<0.2. All other P-values showed no significance and are not indicated. (H, I) RasGRP1-EGFP His 212 variant and WT transfected cells were treated with control, NH4Cl, followed by B cell receptor (BCR) stimulation. Shown are average median levels of P-ERK in EGFP-low expressing cells, corrected for wildtype BCR stimulated cells. WT +BCR was set on 1.0. Shown are means ±SD (BCR, NH4Cl: N = 7, NH4CL + EIPA: N = 4). Wilcoxon test was performed to test differences.

DOI: https://doi.org/10.7554/eLife.29002.008

mode is similar to that seen in SOS:HRas (*Boriack-Sjodin et al., 1998*) and Epac2:Rap1B (*Rehmann et al., 2008*) complexes (*Figure 5E and F*).

## A role for the REM-cdc25 linker in stabilizing the active conformation of RasGRPs

An intriguing aspect of the RasGRP2:Rap1B structure is that the linker connecting the REM domain to the Cdc25 domain (REM-Cdc25 linker) is ordered in its entirety (*Figure 6A and B*). By contrast, this linker is disordered in the RasGRP4 structure (*Figure 6C*). The sequence of the linker is conserved between RasGRP1 and RasGRP2, but is drastically different in RasGRP4 (*Figure 6D*). The structure of the linker as visualized in the RasGRP2 structure is of potential functional importance, because it is inconsistent with adoption of the autoinhibited structure of RasGRP1 that we have defined earlier (*Iwig et al., 2013*) (compare *Figure 6A and B* with *Figure 6E*).

In the structure of autoinhibited RasGRP1, the EF domain is docked on top of helix J of the Cdc25 domain (*Figure 6E*; the helices are labeled as defined for the Cdc25 domain of SOS [*Boriack-Sjodin et al., 1998*]). The docking of the EF domain maintains RasGRP1 in an inactive state in two ways. First, the Cdc25-EF linker physically blocks the active site. Second, in the autoinhibited dimer, interactions between the EF domain of one RasGRP1 molecule and the C1 domain of the other block the membrane-interacting face of the C1 domain (see *Figure 1B* and [*Iwig et al., 2013*]). In the structure of RasGRP2:Rap1B, the REM-Cdc25 linker runs along the surface of helix J, and wraps around it by forming an α helix (*Figure 6A*).

The high sequence conservation of the REM-Cdc25 linker between RasGRP1 and RasGRP2 suggests that the linker might adopt the same conformation in the active states of both molecules. Support for this idea is provided by experiments in which we disrupted ion-pairing hydrogen bonds formed between Arg 131 in the REM-Cdc25 linker and Asp 177 in helix B of the Cdc25 domain, in the RasGRP2:Rap1B structure (*Figure 6B*). We used our cellular assay to test whether mutation of the corresponding residues in RasGRP1, Arg181 and Asp228, has any effect on GEF activity. Upon mutation of the ion-pairing residues, only Arg 181 mutation to alanine (Arg 181 Ala) resulted in somewhat lower levels of basal RasGRP1 signals to P-ERK (*Figure 6F*). These mutations did not significantly impact the RasGRP1-p70S6 kinase pathway (*Figure 6F*), possibly because RasGRP1 signals to S6 are relatively active in the basal state (*Figure 1E*). Both mutants (Arg 181 Ala and Asp 228 Ala) revealed decreased induction of both P-ERK and P-S6 under BCR stimulatory conditions, in comparison to levels induced by wild type RasGRP1 (*Figure 6G*). Taken together, our results indicate that formation of the ion pair between Arg 181 and Asp 228 is necessary for robust activation of RasGRP1.

Another observation that points to the relevance of this linker conformation is that PMA- or antigen receptor-induced phosphorylation of RasGRP1 at Thr 184 in the REM-Cdc25 linker, or RasGRP3 at the corresponding residue (Thr 133), is correlated with increased RasGRP activity. Mutation of these threonine residues to alanine results in reduced, but not eliminated, RasGEF activity (*Aiba et al., 2004*), whereas introduction of a negative charge to mimic a phosphorylated threonine results in higher activity (*Roose et al., 2005*). The corresponding residue in RasGRP2 is Thr 134 (*Figure 6D*) and is phosphorylated in lymphocytes analyzed by total phospho-proteomics (Dr. Doreen Cantrell, set of 11 phospho-peptides from RasGRP2, personal communication). The

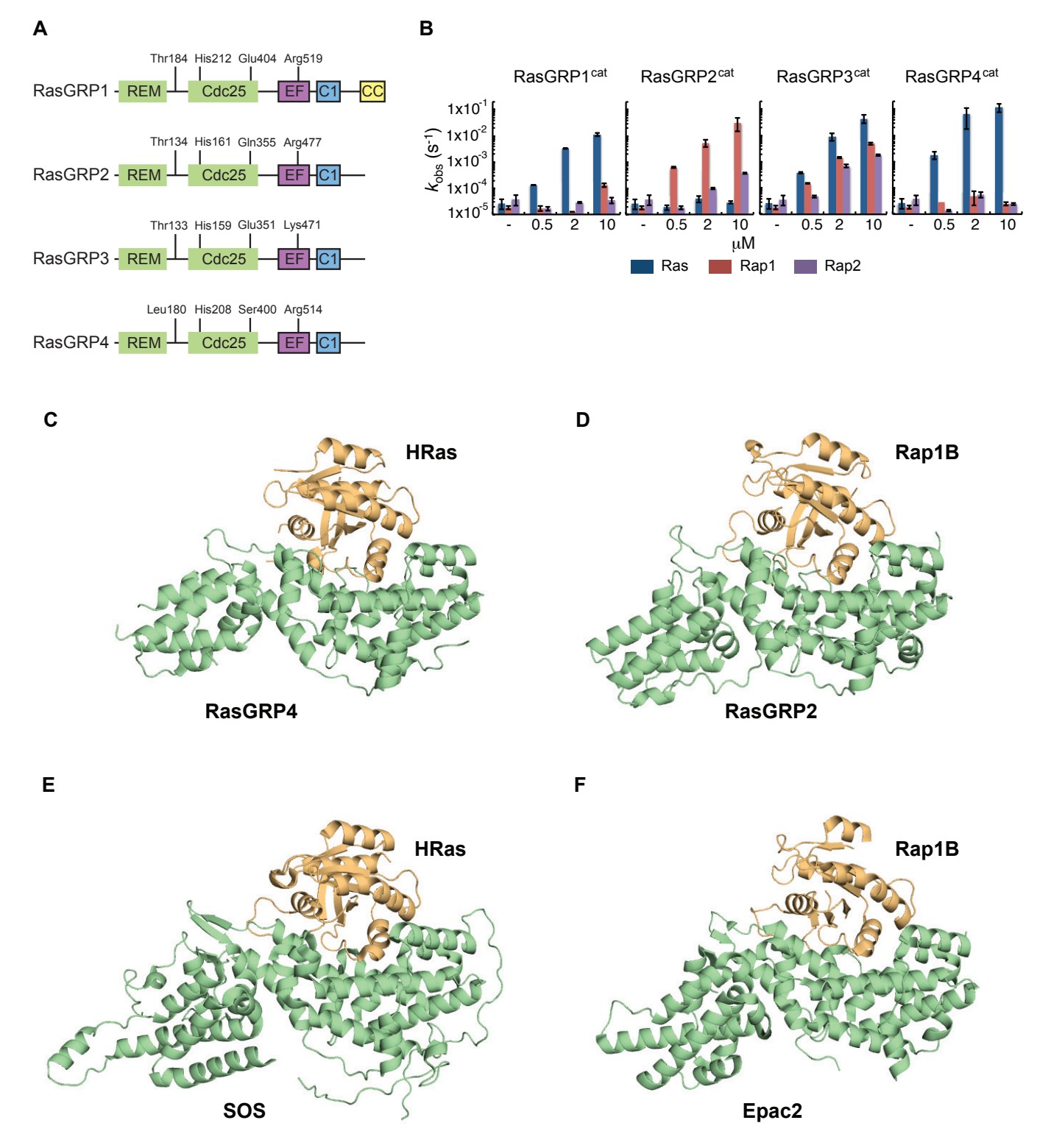

**Figure 5.** RasGRP substrate specificity and active GEF structures. (A) Cartoon showing the four RasGRP isoforms, with key conserved regulatory residues indicated. (B) RasGRP isoform specificity for GTPases HRas, Rap1B, and Rap2A was tested by in vitro nucleotide exchange assays. Shown are the GDP-GTP exchange rates in time for RasGRP-1, -2, -3, and -4 catalytic domains (RasGRPcat, consisting of the REM and Cdc25 domains), in different concentrations (x-axis), for Ras (blue) Rap1B (red), Rap2A (purple). (C–F) Structures of RasGEF (green) – GTPase (orange) complexes are shown with cartoon representation for comparison. RasGRP4:HRas (C) and RasGRP2:Rap1B (D) complex structures were determined by this study. (E) HRas bound

*Figure 5 continued on next page*

*Figure 5 continued*

to the catalytic site of SOS and SOScat domain of the SOScat:HRas complex is extracted from PDB ID:1NVV (*Margarit et al., 2003*). (F) The structures of REM-cdc25 domain of Epac2 and Rap1B in Epac2:Rap1B complex are extracted from PDB ID:3CF6 (*Rehmann et al., 2008*).

DOI: https://doi.org/10.7554/eLife.29002.009

The following source data and figure supplement are available for figure 5:

**Source data 1.** Crystallographic table (RasGRP).
DOI: https://doi.org/10.7554/eLife.29002.011
**Figure supplement 1.** Sequence comparison of RasGRP homologs.
DOI: https://doi.org/10.7554/eLife.29002.010

positioning of the REM-Cdc25 linker in the RasGRP2:Rap1B complexes places the side chain of Thr 134 close to the side chains of the positively charged Lys 172 and Arg 195 in the Cdc25 domain, which provides a possible explanation for the activating effect of phosphorylation of the threonine residue, which adds a negative charge (*Figure 6H*).

## The location and environment of His 212 is consistent with a role as pH sensor

The cellular signaling results with variants of His 212, including the human SNV His 212 Tyr, and the findings that His 212 is a pH sensor, prompted us to investigate how His 212 may impact the transition between the autoinhibited and active states of RasGRP1. Schematic diagrams for these two states are shown in *Figure 7A and B*, which indicate the position of His 212 with respect to the structural elements that rearrange in the transition. His 212 is located within helix A of the Cdc25 domain, at the interface between helices A and J. Helix J is connected to helix K, which forms part of the Ras binding site. In the autoinhibited conformation of RasGRP1, Helix J is part of the platform on which the EF domain is docked (*Figure 7C*). Thus, the location of His 212 is consistent with a role for this residue in the transition between the active and inactive states of RasGRP1. In addition, the structure suggests that the charge on the histidine residue will be important.

An unusual aspect of this histidine is that there are five negatively charged residues in its vicinity (*Figure 7D*). Four of these are provided by helix A (Glu 205, Glu 207, Glu 208 and Glu 211) and the fifth one (Glu 404) is provided by the loop leading into helix J, which packs against helix A (*Figure 7E*). We estimated the pKa values of the histidine sidechains in RasGRP1, using continuum electrostatics as implemented in a web-based server for the program DELPHI (*Sharp and Honig, 1990*; *Wang et al., 2016*). Using the crystal structure of the cdc25 domain of RasGRP1 (*Iwig et al., 2013*), the pKa value of His 212 is calculated to be 6.90, that is, increased by almost 1 pH unit above the pKa value of an isolated histidine (6.0). His 212 has the highest calculated pKa value among the nine histidines in the cdc25, indicating that it is primed to convert from positive to neutral when the intracellular pH increases beyond a neutral value (*Table 1*). The calculated pKa value of this residue is reduced significantly when one or more of the four glutamate sidechains located nearby, in helix A, are substituted by alanine.

We observed a similar cluster of glutamate residues surrounding a His pH sensor in the focal adhesion-associated protein talin, and computational pKa prediction suggested that these glutamate residues have increased pKa values (*Srivastava et al., 2008*). In both RasGRP1 and talin the glutamate network could be part of the pH sensor with histidine. Supporting this idea, experimental work has shown that histidines with coordinating glutamate residues have increased pKa values when determined experimentally by NMR (*Tishmack et al., 1997*; *Hiebler et al., 2017*; *Baran et al., 2008*). As noted earlier, mutation of His 212 to non-titratable residues abrogates pH sensitive activation, which is also consistent with a role for this residue in pH sensing.

We propose that the configuration of residues seen in the autoinhibited RasGRP1 structure is stable when His 212 is protonated, because the concentration of negative charge in this region would favor a positively charged histidine. Neutralization of the histidine may favor disruption of the clustering of negative charge, most likely by movement of Glu 404, in the loop leading into helix J (the other four acidic residues are on the same helix as His 212, and are less likely to move away). In the active structure, the loop bearing Glu 404 packs against the REM-Cdc25 linker (*Figure 7*). In the autoinhibited RasGRP1 structure, the REM-Cdc25 linker positions two hydrophobic sidechains (Leu

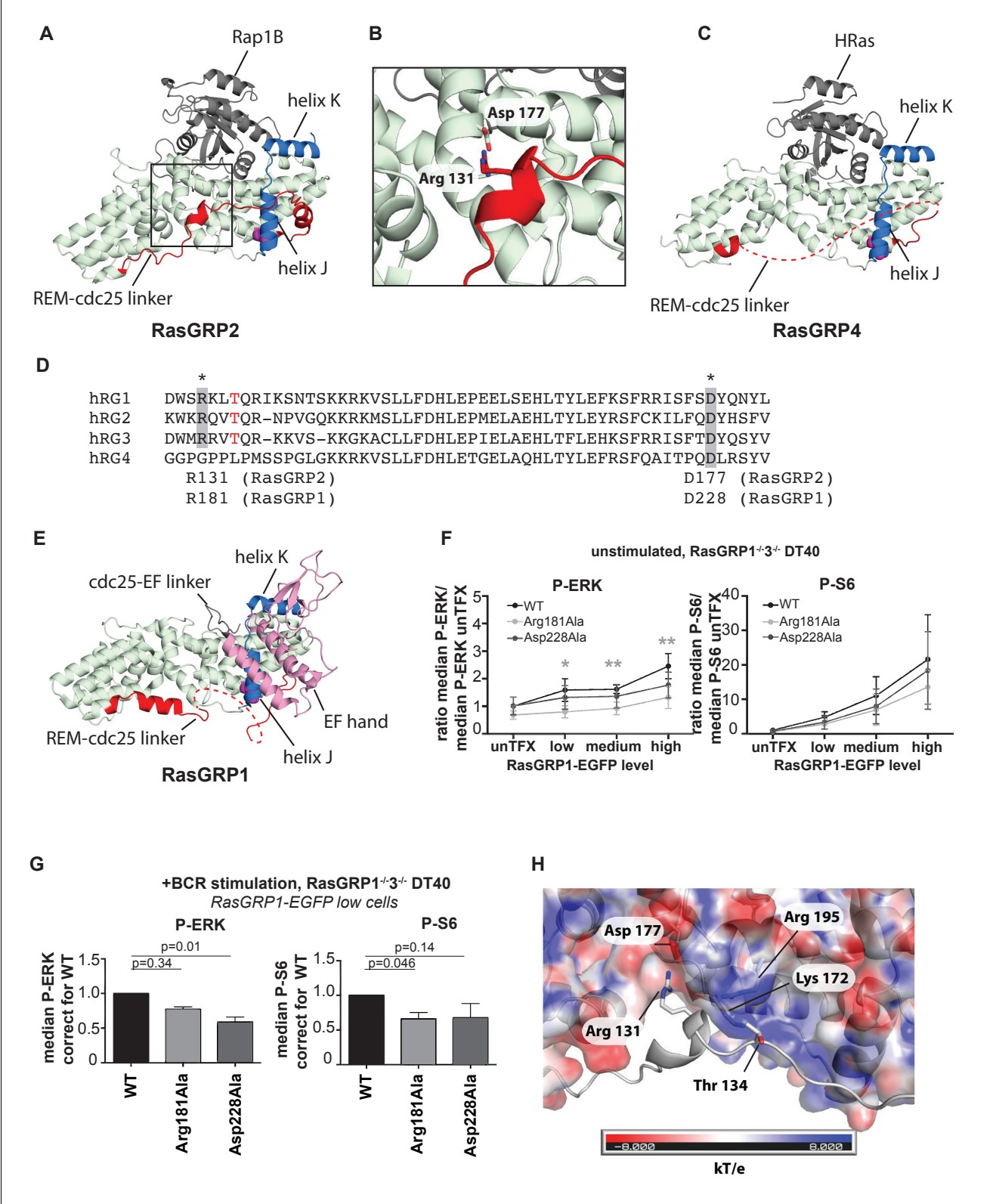

**Figure 6.** A conserved, stabilizing salt-bridge maintains RasGRP1 activity. (**A**) The RasGRP2:Rap1B complex structure is shown with the same orientation as *Figure 5D*. Important structural elements surrounding His 161 are highlighted (REM-cdc25 linker; red, helix J and K; blue, His 161; magenta). (**B**) Close-up view of the area highlighted by a box in *Figure 6A* is shown, and an ion-pair formed by Arg 131 and Asp 177 is shown. (**C**) The RasGRP4:HRas complex structure is shown with the same orientation as *Figure 5C*. The dashed line indicates the disordered Rem-cdc25 linker. His

*Figure 6 continued on next page*

*Figure 6 continued*

208 and other structural elements are highlighted as *Figure 6A*. (D) Alignment of RasGRP isoforms, highlighting RasGRP2's Arg 131 and Asp 177 in grey as conserved residues corresponding to Arg 181 and Asp 228 in RasGRP1. The RasGRP1 phosphorylation site Thr 184 is in red. (E) The RasGRP1 structure (PDB ID: 4L9M [*Iwig et al., 2013*];) is shown with the same orientation as *Figure 6A*. and *Figure 6C*. The structural elements are colored as *Figure 6A and C* with additional elements (cdc25-EF linker; gray, EF hand and C1 domains; pink). (F) DT40 RasGRP1$^{-/-}$RasGRP3$^{-/-}$ cells were transfected with RasGRP1-EGFP WT or variants Arg 181 Ala, Asp 228 Ala. The basal signaling activity of RasGRP1 WT and variants was determined. Shown are average median basal levels of P-ERK and P-S6, corrected for untransfected cells. Kruskal-wallis test was used, with post-test Dunn's multiple comparisons, comparing each mutant to WT. All graphs show mean ±SD (N = 5). *p<0.05, **p<0.01. (G) Cells were unstimulated, or stimulated for 5 min with anti-B cell receptor antibody (clone M4). Only RasGRP1-EGFP low expressing cells were analyzed. Shown are average median levels of P-ERK and P-S6 in EGFP-low expressing cells, corrected for wildtype (WT). WT is set on 1.0. Kruskal-Wallis test was used with post-test Dunn's multiple comparisons, comparing each variant to WT. All graphs show mean ±SD (N = 3), P-values are shown. (H) Electrostatic surface potential of RasGRP2 except the REM-Cdc25 linker, calculated by APBS (*Baker et al., 2001*), is shown. The REM-Cdc25 linker is shown with cartoon model.

DOI: https://doi.org/10.7554/eLife.29002.012

199 and Leu 200) against the hydrophobic interface formed between helix J and the EF domain (*Figure 7C*). Release of the leucine residues, due to destabilization of the conformation of the REM-Cdc25 linker, would weaken the interface with the EF domain and help initiate a transition towards the active conformation. Consistent with this idea, we found that mutation of Glu 404 resulted in altered RasGRP1 activity, while mutation of Glu 208, located within helix A, had no effect. Replacement of Glu 404 by either alanine or arginine both resulted in increased signals to S6 but not to the less basally active RasGRP1-ERK pathway (*Figure 7F and G*, and *Figure 7—figure supplement 1*).

## Concluding remarks

In unstimulated cells, RasGRP1 is in an inactive conformation, in which the Cdc25-EF linker prevents Ras binding to the active site (*Iwig et al., 2013*). In dimeric and autoinhibited RasGRP1, the EF domains from each molecule in a RasGRP1 dimer block the DAG-binding sites on the C1 domains of the dimer partner (*Figure 8*). Lymphocyte receptor stimulation results in increased pHi, increased DAG levels, and increased intracelluar calcium levels, and we propose that these three signals coordinately induce a conformational change in RasGRP1. The structure of the RasGRP2:Rap1B complex that we have now determined establishes an important role for the REM-Cdc25 linker in the transition to the active state. A prominent feature in the active RasGRP2 structure is the formation of an ion-pair between Arg 131, in the REM-Cdc25 linker and Asp 177 in the Cdc25 domain (Arg 181 and Asp 228 in RasGRP1). This salt bridge and the position of the REM-Cdc25 linker reinforces the active conformation of RasGRP and is incompatible with the autoinhibited conformation (*Figure 7*).

The sequence of the REM-Cdc25 linker is conserved between RasGRP1, RasGRP2 and RasGRP3 (*Figure 6D*). This suggests that these proteins share a common regulatory mechanism. For RasGRP1, this mechanism provides the first plausible structural explanation for how phosphorylation of Thr 184 in RasGRP1 (*Zheng et al., 2005*; *Roose et al., 2007*) results in a stable active conformation (*Figure 8*). The sequence of the REM-Cdc25 linker is divergent in RasGRP4, however, and the sequence of the EF domain indicates that RasGRP4 is unlikely to be regulated by calcium (*Iwig et al., 2013*; *Reuther et al., 2002*). Thus it appears that the regulation of RasGRP4 is likely to be different from that of the other three members of the family.

Our analysis of SNPs in the RasGRP proteins led to the finding that His 212 in RasGRP1 functions as a pH sensor. Receptor signaling-induced increases in pHi is expected to convert His 212 to the deprotonated and neutral form, which destabilizes the autoinhibited conformation and potentiates the activation of RasGRP1 by calcium flux and DAG formation at the membrane (*Figure 8*). His 212 is predicted to have an increased pKa value, which is likely influenced by the adjacent cluster of glutamate residues. Supporting this idea, experimental work has shown that histidines with coordinating glutamate residues have upshifted pKa values when determined experimentally by NMR (*Tishmack et al., 1997*; *Hiebler et al., 2017*; *Baran et al., 2008*). At this point we cannot formally rule out the possibility that other histidines in RasGRP1 also function as pH sensors. However, the fact that mutation of His 212 to non-titratable residues abrogates pH sensitive activation, strongly suggests that His 212 is essential for the pH-sensitive function of RasGRP.

Most studies describing pHi changes upon receptor stimulation in lymphocytes have been published roughly 30 years ago (*Cheung et al., 1988*; *Fischer et al., 1988*; *Mills et al., 1985*).

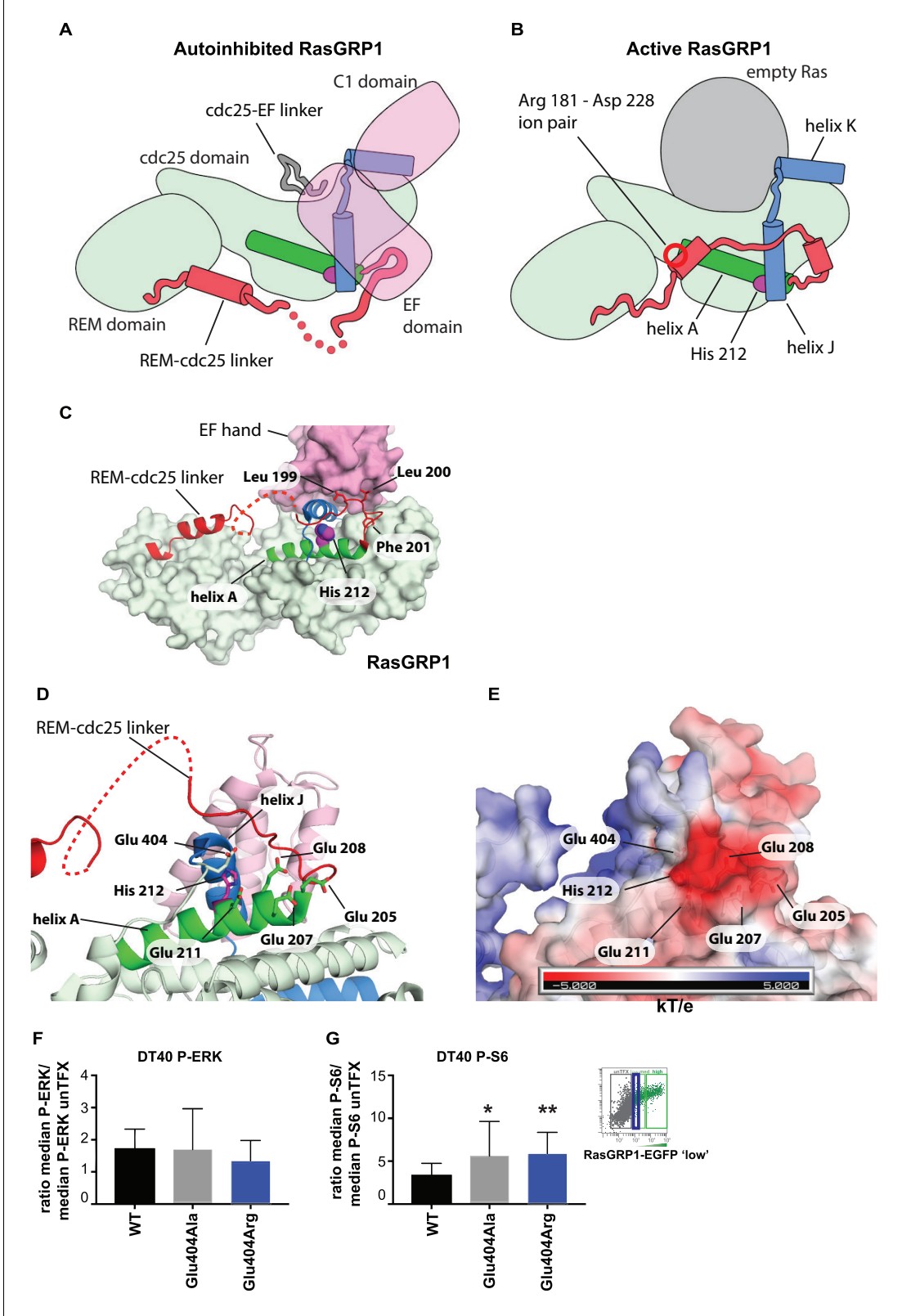

**Figure 7.** His 212 – Glu 404 pair and transition from inactive to active RasGRP1. (A, B) Schematic diagram of RasGRP1 conformations. Color scheme is the same as *Figure 6A,C and E*. In the autoinhibited form (A), the cdc25-EF linker sits in the Ras binding site and the EF domain docks on the cdc25 domain. The active form of RasGRP1 (B) is extrapolated from the active RasGRP2-Rap1B structure. The Rem-cdc25 linker is stabilized by an ion-pair formed by Arg 181 and Asp 228. (C) The interaction between the REM-cdc25 linker and the EF hand is shown. Leu 199 and Leu 200 are stacked on the

*Figure 7 continued on next page*

*Figure 7 continued*

EF hand domain, while Phe 201 is inserted into a hydrophobic pocket in the cdc25 domain. (D) His 212 and surrounding acidic residues in RasGRP1 are shown. (E) Electrostatic surface potential of RasGRP1 is shown with the same view as *Figure 7D*. (F, G) Basal signaling activity of RasGRP1 WT and Glu 404 variants in RasGRP1-/-RasGRP3-/- DT40 cells. Shown are average median levels of P-ERK (F) and P-S6 (G), corrected for wildtype in RasGRP1-EGFP low expressing cells. Friedman test was used, with post-test Dunn's multiple comparisons, comparing each mutant to WT. (N = 6. All graphs show mean ±SD. *p<0.05 and **p<0.01.

DOI: https://doi.org/10.7554/eLife.29002.013

The following figure supplement is available for figure 7:

**Figure supplement 1.** Basal signaling activity of RasGRP1 WT and Glu 404 (**A,B**), or Glu208 (**C,D**) variants in RasGRP1$^{-/-}$RasGRP3$^{-/-}$ DT40 cells.
DOI: https://doi.org/10.7554/eLife.29002.014

Experiments in those studies were performed under different conditions with variable outcomes. The exact effects of receptor stimulation on pHi changes in lymphocytes, and the time scales and subcellular localization of these pHi changes remain largely unknown. Likewise, many questions remain unanswered on the subcellular localization of changes in intracellular calcium and the roles of various ion channels in both resting and stimulated lymphocytes (*Feske et al., 2015*). At this point, tools to measure *local* changes in pHi or calcium levels are not available and how pHi and calcium may regulate signaling proteins locally is not known. Global $Ca^{2+}$ levels in resting cells are in the 20–200 nM range (*Usachev et al., 1995*; *Schwaller, 2010*), although it should be noted that there is an enrichment of calcium ions near the negatively charged polar headgroups of phospholipids in the plasma membrane (*Hille, 2001*). RasGRP1 binds $Ca^{2+}$ with low micromolar affinity (*Iwig et al., 2013*), which suggest that RasGRP1 will not bind $Ca^{2+}$ until the cell is stimulated and calcium levels rise substantially. Debye-Hueckel and Gouy-Chapman models from the early 1900's proposed that ions in an aqueous environment are not evenly distributed. Local entry of calcium through channels in the plasma membrane and local increases in pHi leading to deprotonation of His 212 may provide an interesting interplay in the regulation of RasGRP1. The connection between deprotonation of His 212 and increases in pHi in stimulated lymphocytes we uncovered here motivates us to revisit these studies on pHi in lymphocytes.

More clear is the notion that increased pHi is a hallmark of cancer cells and promotes cancer progression (*Webb et al., 2011*; *White et al., 2017*). For example, in drosophila models, increased pHi is sufficient to induce dysplasia, and it potentiates growth and invasion (*Grillo-Hill et al., 2015*). In the light of cancer, it is of interest to note that RasGRP1 expression levels are increased in T cell leukemia (T-ALL) cells from patients (*Hartzell et al., 2013*). When we biochemically characterized these T-ALL cell lines with an increased level of RasGRP1 expression we noted that RasGRP1 is constitutively recruited to the membrane where it constantly exchanges the nucleotide on Ras (*Ksionda et al., 2016*; *Hartzell et al., 2013*). In agreement, RasGRP1 demonstrates constitutive phosphorylation on Thr 184 in these T-ALL cells (*Ksionda et al., 2016*) arguing that RasGRP1 spontaneously adopts an active conformation. We speculate that an elevated pHi in these cancerous

**Table 1.** Estimated pKa values.

| Residue no. | Estimated pKa values | | | | | |
| --- | --- | --- | --- | --- | --- | --- |
| | wild-type | E208A | E211A | E404A | E205A, E207A, E208A, E211A | E205A, E207A, E208A, E211A, E404A |
| His 212 | 6.90 | 6.81 | 6.71 | 6.71 | 6.44 | 6.12 |
| His 286 | 6.56 | 6.55 | 6.55 | 6.55 | 6.53 | 6.52 |
| His 303 | 6.83 | 6.83 | 6.83 | 6.83 | 6.82 | 6.82 |
| His 315 | 6.17 | 6.16 | 6.16 | 6.17 | 6.15 | 6.15 |
| His 318 | 6.26 | 6.26 | 6.26 | 6.26 | 6.26 | 6.26 |
| His 358 | 6.57 | 6.57 | 6.56 | 6.57 | 6.56 | 6.56 |
| His 381 | 6.27 | 6.27 | 6.27 | 6.27 | 6.27 | 6.27 |
| His 389 | 6.58 | 6.58 | 6.58 | 6.58 | 6.58 | 6.58 |
| His 411 | 6.23 | 6.22 | 6.22 | 6.22 | 6.21 | 6.20 |

DOI: https://doi.org/10.7554/eLife.29002.015

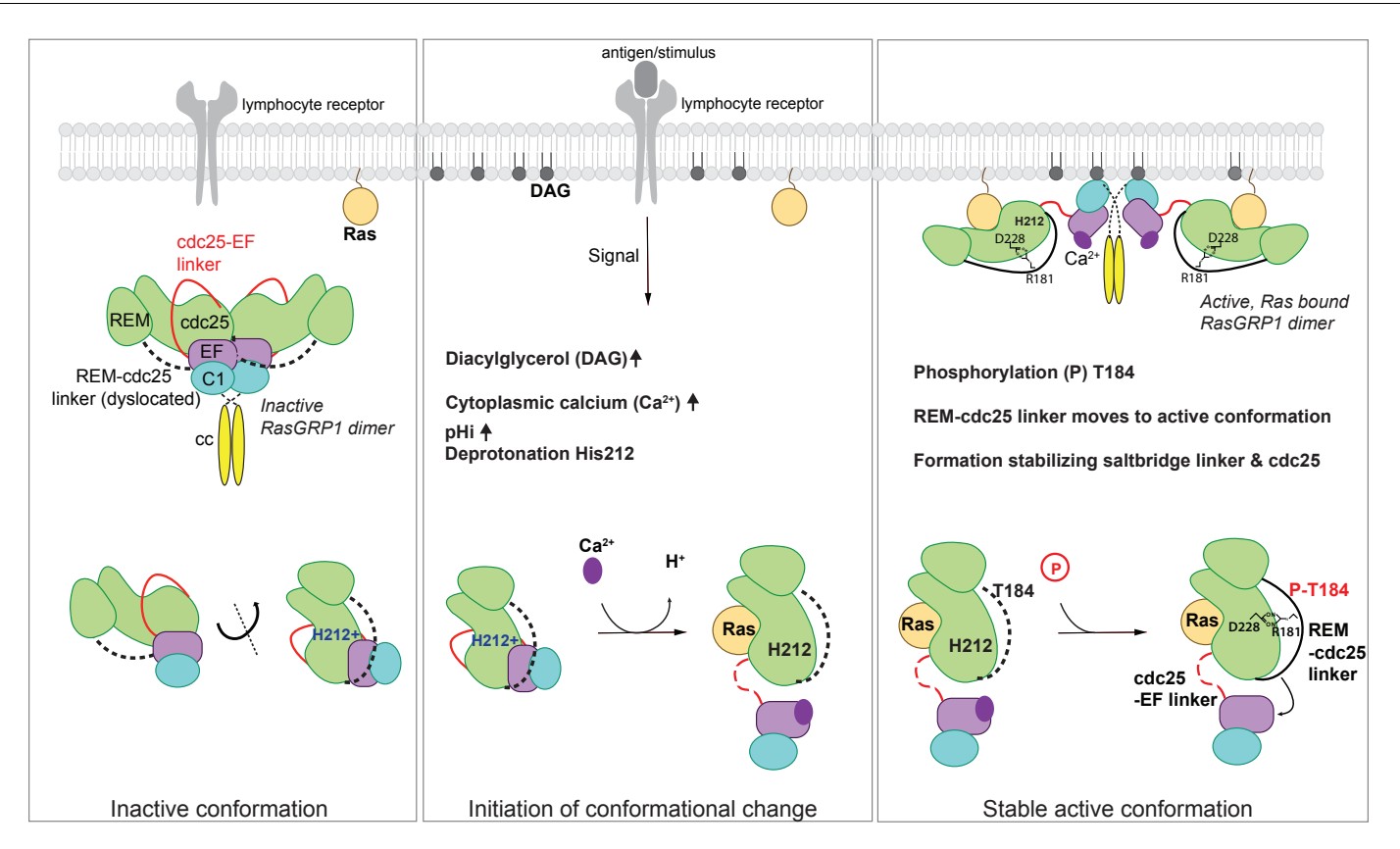

**Figure 8.** Model of RasGRP1 regulation. (Left) In resting cells, RasGRP1's basal activity is autoinhibited through a dimer; the Cdc25-EF linker (red) blocks the Ras binding site, and the EF hands (purple) prevent C1 domains (cyan) from binding diacylglycerol (DAG), and thus block membrane recruitment. (Middle) Upon receptor stimulation, DAG- and intracellular calcium- levels are induced and the intracellular pH (pHi) increases, which we propose leads to deprotonation of His 212, calcium binding to the EF-hand, and initiation of a conformational change that allows for binding to DAG at the membrane. (Right) Formation of a stable, active RasGRP1 conformation. Phosphorylation of Thr 184 promotes the proximity of the REM-Cdc25 linker (black) with the positively charged patch in the Cdc25 domain of RasGRP1. At the same time, a salt-bridge is formed between Asp 228 and R181 to stabilize the active conformation.

DOI: https://doi.org/10.7554/eLife.29002.016

T-ALL cells facilitates opening up of the overexpressed RasGRP1 leading to the constitutively high level exchange activity that we observed (*Ksionda et al., 2016*; *Hartzell et al., 2013*).

# Materials and methods

## Cell lines
Jurkat and low level RasGRP1 expressing JPRM441 cells were previously characterized (*Roose et al., 2005*) and described (*Roose et al., 2007*). RasGRP1$^{-/-}$RasGRP3$^{-/-}$ DT40 cells were described in (*Roose et al., 2007*).

## Medium and buffers
Jurkat and JPRM441 were cultured in RPMI1640 (Hyclone), containing 10% Fetal calf serum (FCS), 1% glutamine, 10 mM Hepes, penicillin and streptomycin. DT40 culture medium contained additional 1% chicken serum. After electroporation, cells were recovered in culture medium without penicillin and streptomycin. Starvation of cells was performed in culture medium containing low FCS (0.2%) for 3 hr, or in plain RPMI1640 for 30 mins. FACS buffer consisted of Phosphate buffered salt (PBS) with 2 mM EDTA, 2% FCS, and 0.1% NaN3.

## Transfection of cells with RasGRP1-EGFP plasmids

RasGRP1$^{-/-}$RasGRP3$^{-/-}$ DT40 cells, or low level RasGRP1 expressing JPRM441 cells (*Roose et al., 2005*) were transfected by electroporation with DNA plasmids encoding EGFPN1-RasGRP1 wildtype or mutant (20 × 10$^6$ cells with 20 microgram DNA)(Biorad Genepulser Xcel). Then cells were rested, stimulated and used for flow cytometry, microscopy, and pHi modulation as described below.

## Flow cytometry assays

The protocol was slightly modified from an earlier published version (*Iwig et al., 2013*). DT40 cells, or JPRM441 cells (*Roose et al., 2005*) were transfected as described above, and recovered in culture medium without antibiotics for 3–4 hr. Cells were washed, resuspended in plain RPMI, seeded 0.4 × 10$^6$ per well in a 96 well round bottom plate, and starved in an incubator for 30 min. DT40 cells were stimulated with anti-IgM (clone M4), cross-linking the B cell receptor, or with vector (RPMI), and JPRM441 cells were treated with RPMI1640 only for 5 min. Cells were fixated in pre-warmed fixations buffer (BD, Cytofix, BD biosciences, San Jose, CA), or in 4% paraformaldehyde in Phosphate buffered saline (PBS), for 15 min at 21°C, washed in FACSbuffer, and permeabilized in either 0.5% Phosflow buffer IV (BD biosciences) for 15 min at 21°C, or MetOH for 30 min on ice. Barcoding protocols were modified from described methods (*Krutzik et al., 2011*). Pacific Blue and Alexa Fluor 750 carboxilic acid succinimidyl-esters (Life Technologies, Grand Island, NY) were added in the methanol or the phosflow buffer IV in titrated serial dilutions, and incubated for 30 min on ice (Methanol), or for 15 min at 21°C (phosflow buffer IV). Cells were washed thoroughly in FACS buffer, and barcoded cells were pooled and incubated for 30 min with antibodies towards P-ERK, P-S6, and for JPRM441 cells we used cleaved caspase or cleaved PARP to exclude (pre-) apoptotic cells from analysis. Cells were washed and analyzed using the LSR II flow cytometer, or LSR Fortessa (BD Biosciences, San Jose, CA, USA). Data were analyzed using Cytobank (Cytobank Inc, Mountain View, CA, USA). In some of the pHi modulation experiments cells were measured without barcoding.

## Antibodies for flow cytometry

Anti-phosphorylated-S6 -PE (clone D57.2.2E, #5316 s, diluted 1:200), anti-phosphorylated ERK-AF647 (clone 137F5, #5376, diluted 1:50), cleaved caspase 3- pacific blue (clone D3E9, #8788, diluted 1:200), or anti-phosphorylated ERK (clone 197G2, #4377 s, diluted 1:50) all from Cell Signaling Technologies, Beverly, MA, USA). Unconjugated P-ERK was followed by AffiniPure F(ab')2 fragment Donkey-anti-Rabbit IgG, conjugated to APC (#711-136-152, diluted 1:50) or PE (#711-116-152, diluted 1:50, Jackson ImmunoResearch, West Grove, PA, USA), or cleaved PARP-AF647 (clone F21-852, #558710, diluted 1:100, BD Biosciences, San Jose, CA, USA)

## Microscopy assays

RasGRP1-RasGRP3 deficient DT40 cells were transfected by electroporation with DNA plasmids encoding EGFPN1-RasGRP1. Cells were cultured for 4 hr, and starved for 30 min in RPMI. Cells were stimulated by anti-IgM (M4), cross-linking the B cell receptor, 20 ng/ml PMA, or with vehicle (RPMI) for 2 min. Cells were fixated in 4% paraformaldehyde in PBS, for 15 min, at 21°C, and washed. Cells were FACS sorted (MoFlo XDP, Beckman Coulter,CA, USA) for RasGRP1-EGFP low or high expression onto poly-L Lysine coated microscopy slides. For each sample 30–50 RasGRP1-EGFP-low cells were captured on a Zeiss confocal microscope, and scored blindly by 2 independent researchers for localization of RasGRP1 (cytoplasmic, or membrane. Cells with partial or full membrane localization (showing a clear line of increased GFP signal at the edge) were counted as 'membrane localization'. Whilst cells only without clear increased membrane fluorescent signal were counted as 'cytoplasma localization'. Profiles were made with Fiji (ImageJ) software.

## pHi modulation and receptor stimulation in cells

Cells were transfected as described above, recovered for 2 hr in culture medium containing 10% FCS without antibiotics, and then transferred to starvation medium containing 0.2% FCS for 3 hr prior to assay. Cells were then were treated with 15 mM ammonium chloride (NH$_4$Cl) in RPMI for 10 min to increase pHi, or pulsed with 15 mM NH$_4$Cl for 15 min, spun down to remove NH$_4$Cl, and resuspended in RPMI with 10 mM EIPA (5'-(N-ethyl-N-isopropyl)amiloride, a selective inhibitor of Na$^+$/H$^+$ exchanger NHE1 activity, Enzo Life Science) for 5 min to decrease pHi (*Choi et al., 2013*).

Next, stimulation with anti-B cell receptor antibody M4 (1:1000) or 20 ng/ml diacylglycerol analogue PMA (phorbol 12-myristate 13-acetate) was performed in presence of $NH_4Cl$, EIPA, or RPMI (control), and after 5 minutes cells were fixated in 4% PFA (paraformaldehyde) in PBS (phosphate buffered salt).

## pHi measurements

$0.5 \times 10^6$ cells per well were plated on a 24 well dishes coated with poly-L Lysine, and induced to be quiescent by maintaining for 3 hr in RPMI with 0.2% FCS. Intracellular pH (pHi) was determined as previously described (*Choi et al., 2013*) in a bicarbonate buffer (25 mM $HCO_3^-$, 115 mM NaCl, 5 mM KCl, 10 mM glucose, 1 mM $KPO_4$, 1 mM $MgSO_4$, and 2 mM $CaCl_2$, pH 7.4) by using cells loaded with 1 µM 2′,7′-bis-(2-carboxyethyl)−5-(and-6)-carboxyfluorescein (BCECF; Invitrogen). To determine steady-state pHi, fluorescence of BCECF at Ex490/Em530 and Ex440/Em530 was acquired every 15 s for 5 min using a plate reader (SpectraMax M5; Molecular Dynamics) and the fluorescence ratios were converted to pHi by calibrating the fluorescence in each well with 10 µM nigericin in 105 mM KCl.

## In vitro nucleotide exchange assay

In vitro nucleotide exchange rates for RasGRP1-4 proteins (RasGRP1cat, RasGRP2cat, RasGRP3, and RasGRP4cat) and small GTPases (HRas, Rap1B, and Rap2A) were measured and analyzed as described previously (*Iwig et al., 2013*), except that the final GTPase concentration was 1 µM instead of 500 nM.

## Protein expression and purification

RasGRP2cat (residues 1-394), RasGRP2cat (1-425), and RasGRP4cat (46-460) expression vectors were constructed by inserting the genes into 2CT-10 plasmid (attaches an N-terminal hexa-histidine tag followed by MBP and TEV protease cleavage site), a gift from Scott Gradia (Addgene plasmid # 55209). RasGRP3cat (1-418) was cloned into pSMT3 vector. Rap1B (1-167), Rap1B (1-175), and Rap2A (1-167) was cloned into pProEX HTb vector (Invitrogen). All proteins correspond to human genes.

Non-labeled RasGRPcat domains were expressed in T7 Xpress cells (NEB) in Terrific Broth media. RasGRP2cat labeled with selenomethionine (SeMet) (residues 1–394) was expressed in BL21(DE3) cells in M9 minimal media with 50 µg/ml of each amino acid except methionine, 5 µg/ml methionine and 50 µg/ml SeMet.

For RasGRPcat domains, the cells were grown at 37°C with 100 µg/ml ampicillin, and induced with 0.25 mM IPTG at 15°C for 14–18 hr. The cells were suspended in the Ni-A buffer (25 mM Tris pH8.0 500 mM NaCl, 10% Glycerol, 20 mM imidazole, and 5 mM β-mercaptoethanol) supplemented with 200 µM AEBSF, 5 µM leupeptin and 500 µM benzamidine, and lysed by a cell disrupter. The cell lysate was cleared by ultracentrifugation with 142,032 x g for 1 hr and applied to HisTrap column (GE Healthcare). The protein was eluted by Ni-A buffer supplemented with 500 mM imidazole. The MBP tag was cleaved off by addition of TEV protease while dialyzing against the buffer (25 mM Tris pH8.0, 100 mM NaCl, 10% glycerol, 1 mM TCEP). The protein was run through HisTrap column and the flowthrough was collected. The protein was concentrated using Amicon Ultra ultrafiltration unit (EMD Millipore) and run on an S-75 size-exclusion column equilibrated with SEC buffer (25 mM Tris pH 8.0, 100 mM NaCl, 10% glycerol, 1 mM TCEP). The protein was concentrated and stored at −80°C after flash-freezing using liquid nitrogen.

HRas (residues 1–166), Rap1B (1-167), Rap1B (1-175), and Rap2A (1-167) were expressed and purified as described previously for HRas (*Iwig et al., 2013*).

## Crystallization and structure determination

The RasGRP2:Rap1B complex and the RasGRP4:HRas complex were formed by mixing the proteins at 1:2 ratio of the GEF and either Rap1B or HRas. After addition of alkaline phosphatase (SIGMA, P0114), the sample was incubated at 4°C overnight. The complexes were purified by gel filtration (Superdex 75 column) in SEC buffer.

Crystals of SeMet-labeled RasGRP2cat bound to Rap1B were obtained by sitting-drop vapor diffusion (500 µL reservoir volume) by mixing 2 µL of protein (30 mg/mL) with 2 µL of the reservoir

solution containing 17% PEG3350 and 10 mM sodium citrate (pH 5.6) at 20°C. The crystals were streak-seeded from another crystallization drop using a cat whisker 3 min after mixing the drops. Crystals were harvested for data collection after one month, cryoprotected by incubating them in mother liquor supplemented with 10% xylitol, and flash-frozen in liquid nitrogen. Most of the crystals grown in this condition were in space group $P2_122_1$, and exhibited anisotropic diffraction, but the best data were obtained from one crystal with space group $I2_12_12_1$. All X-ray diffraction data were collected at Advanced Light Source beamlines 8.3.1 and 8.3.2. The data were integrated and scaled using XDS (*Kabsch, 2010*) and Aimless (*Evans, 2006*).

Crystals of the RasGRP4cat:HRas complex were obtained by hanging-drop vapor diffusion (500 μL reservoir volume) by mixing 1 μL of protein (10 mg/mL) with 1 μL of the reservoir solution containing 20% PEG3350 and 300 mM sodium thiocyanate at 20°C. The crystals were cryoprotected by the reservoir solution supplemented with 20% glycerol before flash-frozen in liquid nitrogen. The space group of the RasGRP4cat:HRas crystals was initially assigned as $P6_322$, and molecular replacement using Phaser (*McCoy et al., 2007*) with the RasGRP1 Cdc25 domain (PDB ID: 4L9M) as a search model placed three copies of RasGRP4cat in the asymmetric unit. We were able to place HRas bound to RasGRP4cat as observed in the SOS-HRas complex (*Boriack-Sjodin et al., 1998*), but we could not model the REM domain due to weak electron density. We then processed the data in a lower-symmetry space group ($P3_212$), The final model contains six copies of the RasGRP4cat:HRas complex. The model was rebuilt using Coot (*Emsley et al., 2010*) and the structure was refined by Refmac (*Murshudov et al., 2011*) and Phenix (*Adams et al., 2010*) in an iterative manner (*Figure 5— source data 1*). The application of a twinning operator ($h, k, l \rightarrow k, h, -l$) gave better electron density for the REM domain.

Anomalous diffraction data for crystals of SeMet-labeled RasGRP2cat bound to Rap1B were integrated and scaled as for the RasGRP4cat-HRas data. Phase were determined by molecular replacement using Phaser with RasGRP4cat-HRas as a search model, and refinement was performed with Refmac and Phenix using anomalous data to reduce the model bias with rebuilding the model using Coot (*Figure 5—source data 1*).

## Acknowledgements

The authors thank Marsilius Mues and Andre Limnander for assistance with mutagenesis, Jesse E Jun for advice on DT40 cell assays, Frank L Bos for technical assistance with confocal microscopy, Tiago Barros and Christine Gee for technical assistance with crystallography, and members of the Roose, Kuriyan and Arthur Weiss labs for helpful discussions. The Rap1 and Rap2 plasmids were gifts from Dr. Xuewu Zhang at UT Southwestern.

This research was supported by a P01 Program grant from NIH-NIAID (AI091580 – to JK and JPR). Further support came from a Marie Curie International Outgoing Fellowship # PIOF-GA-2012–328666 (to YV) and R01-AI104789 (to JPR), NIH grant 5F32GM095149-03 (JSI), NIH grants R01-CA197855 (DLB) and F32-CA177085 (KAW) and the Dutch Arthritis Foundation, the Foundation for Scientific Research in Internal Medicine (SWOIG) Nijmegen, the KWF (Dutch Cancer Society), the Nijbakker-Morra Foundation, and the Nora Baart Foundation (all to ABJ).

## Additional information

### Competing interests

John Kuriyan: Senior editor, *eLife*. The other authors declare that no competing interests exist.

### Funding

| Funder | Author |
| --- | --- |
| National Institute of Allergy and Infectious Diseases | John Kuriyan<br>Jeroen P Roose |
| National Cancer Institute | Katharine A White<br>Diane L Barber |

| | |
|---|---|
| Marie Curie International Out-going Fellowship | Yvonne Vercoulen |

The funders had no role in study design, data collection and interpretation, or the decision to submit the work for publication.

## Author contributions

Yvonne Vercoulen, Conceptualization, Data curation, Formal analysis, Funding acquisition, Investigation, Methodology, Writing—original draft; Yasushi Kondo, Jeffrey S Iwig, Conceptualization, Data curation, Formal analysis, Investigation, Methodology, Writing—review and editing; Axel B Janssen, Formal analysis, Investigation; Katharine A White, Diane L Barber, Formal analysis, Investigation, Writing—review and editing; Mojtaba Amini, Investigation; John Kuriyan, Jeroen P Roose, Conceptualization, Supervision, Funding acquisition, Writing—original draft, Writing—review and editing

## Author ORCIDs

Yvonne Vercoulen http://orcid.org/0000-0002-5060-2603
Yasushi Kondo http://orcid.org/0000-0003-2131-575X
Axel B Janssen http://orcid.org/0000-0002-9865-447X
Diane L Barber http://orcid.org/0000-0001-7185-9435
John Kuriyan https://orcid.org/0000-0002-4414-5477
Jeroen P Roose https://orcid.org/0000-0003-4746-2811

## Decision letter and Author response

Decision letter https://doi.org/10.7554/eLife.29002.021
Author response https://doi.org/10.7554/eLife.29002.022

## Additional files

### Major datasets

The following datasets were generated:

| Author(s) | Year | Dataset title | Dataset URL | Database, license, and accessibility information |
|---|---|---|---|---|
| Yasushi Kondo, Jeffrey S Iwig, John Kuriyan | 2017 | Structure of RasGRP2 in complex with Rap1B | http://www.pdb.org/pdb/search/structid-Search.do?structureId=6AXF | Publicly available at PDB (accession no: 6AXF) |
| Yasushi Kondo, Jeffrey S Iwig, John Kuriyan | 2017 | Structure of RasGRP4 in complex with HRas | http://www.pdb.org/pdb/search/structid-Search.do?structureId=6AXG | Publicly available at PDB (accession no: 6AXG) |

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
