## [Decision Letter]

Thank you for submitting your article "A Histidine pH Sensor Regulates Activation of the Ras-specificGuanine Nucleotide Exchange Factor RasGRP1" for consideration by *eLife*. Your article has been reviewed by three peer reviewers, one of whom, Volker Dötsch (Reviewer #1), is a member of our Board of Reviewing Editors, and the evaluation has been overseen by Philip Cole as the Senior Editor. The following individual involved in review of your submission has agreed to reveal their identity: David Lambright (Reviewer #2).

The reviewers have discussed the reviews with one another and the Reviewing Editor has drafted this decision to help you prepare a revised submission. As you see, all reviewers see the importance of this study. However, with respect to the significance of some of the measured effects questions were raised. One particular topic that was discussed among the reviewers is the issue of the pKa of His212 and effects of local pH variations within the cell. The comments were:

1) What effect does the local environment (e.g. glutamate cluster, solvent exposure) have on the pKa of HIs 212? Is its pKa likely to be different from the usual solvent-exposed value (ca. 6, which is well below the pH range manipulated in the cell-based experiments) and, if so, how might this relate to the proposed pH-sensor mechanism?

2). Related to the question above: do other histidine residues potentially contribute to the observed pH effect?

3) The intracellular pH can be quite different from the bulk intracellular pH depending on the organelle and location. For example, the nucleus is thought to have a higher pH. Likewise it is discussed that the high amount of negative charge at the membrane results in a locally lower pH by having a locally higher H^+^ concentration according to Debye-Hueckel and Gouy-Chapman models. This could inhibit activation under normal conditions. However, release of calcium might replace the H^+^ near the membrane, resulting in the required rise in pH. Calcium release might therefore have a dual role. It is difficult to measure such localized pH values but such effects should be discussed.

In principle all the reviewers think that determining the pKa of His212 and investigating the effect of local pH values would be important to validate the proposed mechanism. However, the reviewers also realize that this would be very difficult or even impossible with the techniques currently available. Nevertheless the reviewers think that these issues should be discussed and the proposed model be adapted.

Summary:

RasGRPs are highly regulated GEFs that exist in 'inactive' and 'active' conformations. Activation of RasGRPs requires release of autoinhibitory interactions, including the rearrangement of an inter-domain linker. Single nucleotide polymorphisms (SNP) are a common genetic variation, giving rise to differences in single nucleotide changes. In this study, the authors analyzed RasGRP1 SNP variants and identified a conserved histidine (212) in the Cdc25 catalytic domain that appears critical for RasGRP1 autoinhibition. Biochemical and cell biology experiments are conducted to evaluate whether the activity of RasGRP1 is sensitive to cellular pH via modulation of the charge state of His 212. The previously published structure of autoinhibited RasGRP1 combined with new X-ray structures of active RasGRP4 complexed to HRas and RasGRP2 complexed to Rap1b, aided in delineating the role of this histidine in RASGRP1 autoinhibition in vitro and in cells. Based on results from these studies, the authors propose that His212 acts as a pH sensor that controls conformational changes important for activation of RasGRP1.

Essential revisions:

1) In Figure 1, the y-axes for panels E and F are ambiguous, and it is unclear what is being measured. Is this a ratio of phospho-signal relative to untransfected readings? Or an absolute measurement/count based on flow cytometry? If a ratio, why are the WT signals not starting at a 1.0 ratio? Additionally, panel E would suggest much higher signal for p-S6 over p-ERK, yet panel F shows a range of 0-25 for p-ERK but only 0-3 for p-S6. Was a Western blot done on cell lysates to confirm that the flow cytometry signals correspond increased p-ERK and p-S6?

2) In Figure 2, it is not convincing that H212T/A mutants show increased signaling over WT, even at "high" expression levels.

3) In Figure 3, the opposite trend in signaling induction from Figure 1 is observed, in which p-ERK levels increase 7-9 fold, yet p-S6 only increases 2-3 fold. Some discussion is needed here, considering that p-ERK was indicated as being less potent for this system (Figure 1). Further, the shift observed in the flow signal for p-ERK seems very large for just an increased induction of MAPK signaling, perhaps indicating non-specific effects. Western blot analysis may help to clarify this concern. The range of changes seen in Figure 3 are very modest at best. What do these signaling outputs look like in cells expressing "high" levels of RasGRP1?

Images shown in Figure 3 indicating RasGRP localization are not of high quality, so it remains questionable whether there is potent membrane localization. Were these images acquired on "low" expressing RasGRP1 cells? If so, the expression already seems extremely high. Matching images are not provided for Figure 3. It is unclear how the binning is done for plasma membrane vs. cytoplasmic RasGRP. Supporting images are needed.

4) The importance of ion pairs is typically analyzed by mutating both amino acids individually (as the authors have done) and by a charge swap experiment in which the two amino acids are exchanged in a double mutant (with the expectation that the active is not changed compared to wild type). Has such a charge swap experiment been done?

5) In Figure 4, why was NH4Cl dosage used as a standard for increasing pHi over PMA + ionomycin treatment? As intracellular pH lowering can lead to induction of cell death pathways, it is unclear why markers of cell death/apoptosis were not used, especially given the flow cytometry-based platform that is in place. While much of the data in this paper do not show statistically significant changes, panels 4H and 4I do show increases and decreases respectively across H212 mutation status -/+ EIPA pre-treatment.

Figure 4 may be more suitable for supplemental materials.

6) Figure 5. The SNVs results are reported for the RasGRP1 isoform. Are structures for autoinhibited states of RasGRP2 or RasGRP4 available that can be compared to RasGRP1, or are the assumptions/predictions primarily made off of sequence homology? While a paper is cited to address the high similarity between different RasGRP isoforms, a careful sequence alignment and comparison of RasGRPs would be helpful.

7) Given the data for the phospho-mimetic mutation at T134 in RasGRP1, was this mutation made in the RasGRP2/4 GEFs being crystallized in this paper?

8) In Figure 7, the changes in phospho-protein levels do not appear to significantly change with E404A/R mutation. Further, why just E404? Based on earlier statements, mutation-scanning at the other Glu residues mentioned should be considered. Have such mutations been investigated?

Regarding the negatively charged patch (Figure 7), E211 may have significant contribution considering its positioning before H212.

---

## [Author Response]

The comments were:1) What effect does the local environment (e.g. glutamate cluster, solvent exposure) have on the pKa of HIs 212? Is its pKa likely to be different from the usual solvent-exposed value (ca. 6, which is well below the pH range manipulated in the cell-based experiments) and, if so, how might this relate to the proposed pH-sensor mechanism?

This is an excellent point. In the original version we already made the comparison between RasGRP1 and talin, which both have clusters of glutamate around the Histidine pH sensor.

Since RasGRP1CEC construct has molecular weight of 64 kDa, and 18 histidine residues, NMR approaches would be too ambitious to take. Instead, we used a web server to calculate the estimated pKa of His212 in RasGRP1. We estimated the pKa values of the histidine sidechains in RasGRP1, using continuum electrostatics as implemented in a web-based server for the program DELPHI. Using the crystal structure of the cdc25 domain of RasGRP1, the pKa value of His 212 is calculated to be 6.90, i.e., increase by almost 1 pH unit above the pKa value of an isolated histidine (6.0). His 212 has the highest calculated pKa value among the nine histidines in the cdc25 (Table 1). Furthermore, the pKa value of His 212 is reduced significantly when one or more of the four glutamate sidechains located nearby, in helix A, are substituted by alanine.

We have now included this table as Table 1. These analyses support the notion that protonation state of His212 in RasGRP1 is sensitive to pH change.

2). Related to the question above: do other histidine residues potentially contribute to the observed pH effect?

We cannot formally rule out the possibility that other histidines in Rasgrp also function as a pH sensor. However, His 212 has the highest calculated pKa value among the nine histidines in the cdc25, indicating that it is primed to convert from positive to neutral when the intracellular pH increases beyond a neutral value (Table 1).

Furthermore, the cellular experiments in Figure 4 that show that mutation of His212 to non-titratable residues abrogates pH sensitive activation combined with the newly added table demonstrating that His212 likely has an upshifted pKa, strongly suggest that His 212 is the major pH sensor and alterations in pH do not lead to Rasgrp1-dependent signaling effects when His 212 is mutated. We added a few sentences regarding this point to the Discussion.

3) The intracellular pH can be quite different from the bulk intracellular pH depending on the organelle and location. For example, the nucleus is thought to have a higher pH. Likewise it is discussed that the high amount of negative charge at the membrane results in a locally lower pH by having a locally higher H^+^ concentration according to Debye-Hueckel and Gouy-Chapman models. This could inhibit activation under normal conditions. However, release of calcium might replace the H^+^ near the membrane, resulting in the required rise in pH. Calcium release might therefore have a dual role. It is difficult to measure such localized pH values but such effects should be discussed.In principle all the reviewers think that determining the pKa of His212 and investigating the effect of local pH values would be important to validate the proposed mechanism. However, the reviewers also realize that this would be very difficult or even impossible with the techniques currently available. Nevertheless the reviewers think that these issues should be discussed and the proposed model be adapted.

We appreciate that the reviewers point out that current technology does not yet allow for accurate measurements of local pH values.

The model of locally higher proton concentration and locally lower pH caused by the high amount of negative charge at the inner leaflet of the plasma membrane is an interesting point. Indeed, influx of calcium may replace the protons and serve a dual role in activating RasGRP1 by (i) binding to the EF hands and (ii) resulting in a local increase in pH. It’s also been postulated that there is an enrichment of calcium ions near the negatively charged polar headgroups of phospholipids in the plasma membrane. This is however all very speculative and we have no data to support these ideas.

We have added a more neutral paragraph to the Discussion that it will be of interest in the future to understand the balance between protons, local pH, and local influx of calcium ions and what impact these make in the regulation of RasGRP1 (and other signaling molecules).

Essential revisions:1) In Figure 1, the y-axes for panels E and F are ambiguous, and it is unclear what is being measured. Is this a ratio of phospho-signal relative to untransfected readings? Or an absolute measurement/count based on flow cytometry? If a ratio, why are the WT signals not starting at a 1.0 ratio? Additionally, panel E would suggest much higher signal for p-S6 over p-ERK, yet panel F shows a range of 0-25 for p-ERK but only 0-3 for p-S6. Was a Western blot done on cell lysates to confirm that the flow cytometry signals correspond increased p-ERK and p-S6?

The Y-Axes for panels 1E and F indicate the ratio of the median phospho-signal for cells in each gate relative to the median phospho-signal of cells in the untransfected gate in the WT transfection sample. This has been used to normalize for signal variation between independent experiments. This explanation has now been added to the figure legend, and the axis labels have been clarified as ‘ratio median PERK/

median P-ERK unTFX’.

The signals start at a 1.0 ratio for the untransfected gate, as in Figure 1. In Figure 1 we depict the cells from the ‘low’ transfection gate, thus the ratios are >1.0. To clarify, a small figure indicating the ‘low’ gate was added to panel 1F.

Indeed, the P-S6 signal is much higher than P-ERK. The axes numbering was swapped in 1F. We apologize for the mistake, this has now been corrected.

2) In Figure 2, it is not convincing that H212T/A mutants show increased signaling over WT, even at "high" expression levels.

In Figure 2 we show the range of expression levels and the changes in signaling by depicting the ratios over cells in the unTFX gate of the WT condition to equalize the baseline. We see clear differences and this is supported by statistical tests (asterisks indicate P<0.05 and smaller).

Error bars (standard deviation) and the display of all different expression levels in one figure may have obscured the findings. To clarify this, we now also show the ratio of P-S6 and P-ERK for each mutant versus wild type for each gate (low, medium, high) separately and we added the data as Figure 2—figure supplement 1.

3) In Figure 3, the opposite trend in signaling induction from Figure 1 is observed, in which p-ERK levels increase 7-9 fold, yet p-S6 only increases 2-3 fold. Some discussion is needed here, considering that p-ERK was indicated as being less potent for this system (Figure 1). Further, the shift observed in the flow signal for p-ERK seems very large for just an increased induction of MAPK signaling, perhaps indicating non-specific effects.

In the original manuscript we stated:

“Our quantitative flow cytometric analyses revealed that RasGRP1 signals strongly to P-S6 in the basal state; basal signals from RasGRP1 to ERK do occur but are more modest (Figure 1). Upon BCR stimulation with M4 antibody, we typically observe induced levels of P-ERK and P-S6 that are 7-9 fold and 2-3 fold over baseline, respectively (Figure 3). Note that induction of ERK phosphorylation is more robust than S6 phosphorylation upon BCR stimulation.”

The differences in the robustness of mTOR-S6 versus Ras-ERK signaling depends on whether the basal state is examined or whether cells are stimulated. We now added the following sentence to clarify this better:

“Thus S6 phosphorylation appears a more robust event in the basal state (Figure 1) and ERK phosphorylation is a more robustly triggered kinase pathway in stimulated cells.”

Western blot analysis may help to clarify this concern. The range of changes seen in Figure 3 are very modest at best. What do these signaling outputs look like in cells expressing "high" levels of RasGRP1?

We have seen the same robust basal mTOR-S6 signals and exponential (digital) Ras-ERK signaling by western blots in published work (Das et al., Cell 2009, Daley et al., 2013). We did not perform western blot assays here because it does not allow for gating on the “low”, “medium” and “high” RasGRP1 expression that we do for the phospho-flow assays.

Images shown in Figure 3 indicating RasGRP localization are not of high quality, so it remains questionable whether there is potent membrane localization. Were these images acquired on "low" expressing RasGRP1 cells? If so, the expression already seems extremely high. Matching images are not provided for Figure 3. It is unclear how the binning is done for plasma membrane vs. cytoplasmic RasGRP. Supporting images are needed.

We inserted a higher resolution figure in 3F.

Furthermore, we added Figure 3—figure supplement 1 showing high-resolution, representative images for each condition in 3G. The plasma membrane localization is evident by increased signal intensity at the edge of the cells (example Figure 3).

Scores were performed as described before in Daley et al., 2013: Blind scoring by 2 independent researchers was performed to count all single cells that were sorted and imaged from the ‘low’ expressing gates. Cells with partial or full membrane localization (showing a clear line of increased GFP signal at the edge) were counted as ‘membrane localization’. Whilst cells without clear increased membrane fluorescent signal were counted as ‘cytoplasma localization’. We now added this information to the Materials and methods section.

4) The importance of ion pairs is typically analyzed by mutating both amino acids individually (as the authors have done) and by a charge swap experiment in which the two amino acids are exchanged in a double mutant (with the expectation that the active is not changed compared to wild type). Has such a charge swap experiment been done?

We assume the reviewer is pointing to Figure 5 here, where R181A and D228A mutants were shown. The charge-swap experiment has not been done. In addition, because of technical reasons, we are somewhat limited with our in vitro nucleotide exchange experiments (See described below, related to the T184 phosphorylation site).

5) In Figure 4, why was NH4Cl dosage used as a standard for increasing pHi over PMA + ionomycin treatment?

NH4Cl addition to cells is a standardized method to increase pHi. Our goal was to test effects of pHi changes on RasGRP1 signaling when we provide a subsequent B cell receptor stimulus. Therefore, we did not use PMA/ionomycin because this already induces RasGRP1 membrane localization and calcium flux. We have now better explained this in the text.

As intracellular pH lowering can lead to induction of cell death pathways, it is unclear why markers of cell death/apoptosis were not used, especially given the flow cytometry-based platform that is in place. While much of the data in this paper do not show statistically significant changes, panels 4H and 4I do show increases and decreases respectively across H212 mutation status -/+ EIPA pre-treatment.

Regarding the point of apoptosis, lowering of intracellular pHi has been shown to induce cell death pathways (we cited this in the original version), but experimental induction of cell death requires pHi <7.0 in the absence of any other apoptotic cues. With EIPA treatment in our assays, we observe mean pHi values of 7.2-7.3, with no single EIPA treatment replicate dropping below 7.1. We now state this more clearly.

Nevertheless, dead fragmented cells were always gated out of the analysis using forward and side scatter plots. We attempted to stain for (early) apoptotic cells with anti-cleaved parp and anti-cleaved caspase antibodies, but these stainings did not work DT40 cells in conjunction with the phosphor-signaling stains.

Figure 4 may be more suitable for supplemental materials.

We believe Figure 4 will help readers (such as Immunologists and cancer biologists) who are not so familiar with intracellular pH changes and potential effects via His protonation.

6) Figure 5. The SNVs results are reported for the RasGRP1 isoform. Are structures for autoinhibited states of RasGRP2 or RasGRP4 available that can be compared to RasGRP1, or are the assumptions/predictions primarily made off of sequence homology?

To our best knowledge, there is neither RasGRP2 nor RasGRP4 autoinhibited structures reported. However, the crucial residue (W454 in RasGRP1) to block the Ras/Rap binding pocket of the cdc25 domain is highly conserved among RasGRP proteins (Iwig et al., 2013, Figure 5). Therefore, we believe the autoinhibited structures are common in RasGRP proteins.

While a paper is cited to address the high similarity between different RasGRP isoforms, a careful sequence alignment and comparison of RasGRPs would be helpful.

We have now added a sequence alignment to Figure 5—figure supplement 1.

7) Given the data for the phospho-mimetic mutation at T134 in RasGRP1, was this mutation made in the RasGRP2/4 GEFs being crystallized in this paper?

The residue reported to be phosphorylated in RasGRP1 is T184. We have prepared T184E mutant with RasGRP1CEC construct (the same construct as crystallized in Iwig et al., 2013), comprising REMcdc25- EF-C1 domains and takes autoinhibited conformation. We attempted to see the activating effect of the mutation with in vitro nucleotide exchange experiments. However, we could not see clear difference in the activities between wildtype and the mutant. The lack of clear phenotype is possibly caused by our truncated construct of RasGRP1, but we have not been able to prepare full-length proteins of RasGRP1 to test the activity yet. Also, we have not been able to produce RasGRP2 and RasGRP4 CEC constructs. Because of these technical problems, we have not been able to test the phospho-mimetic mutation in RasGRP2 and RasGRP4.

8) In Figure 7, the changes in phospho-protein levels do not appear to significantly change with E404A/R mutation. Further, why just E404? Based on earlier statements, mutation-scanning at the other Glu residues mentioned should be considered. Have such mutations been investigated?

In Figure 7 (P-ERK) indeed there were no significant changes, possibly due to the fact that basal PERK is very low in these cells in general (Figure 1). However, in Figure 7 we see significantly increased levels of phosphorylated S6, indicated by asterisks. To improve clarity, we now show the ratios for only RasGRP1-EGFP low expressing cells, and moved the current figures to Figure 7 supplement (A and B). We also tested Glu208 mutants. However, this did not show any significant changes. Graphs were added to the Figure 7—figure supplement 1(C and D).

Regarding the negatively charged patch (Figure 7), E211 may have significant contribution considering its positioning before H212.

This point is addressed in Table 1. The pKa value of His 212 is reduced significantly when one or more of the four glutamate sidechains located nearby, in helix A, are substituted by alanine.